# Probing CP symmetry and weak phases with entangled double-strange baryons

The BESIII Collaboration*✉

Though immensely successful, the standard model of particle physics does not offer any explanation as to why our Universe contains so much more matter than antimatter. A key to a dynamically generated matter–antimatter asymmetry is the existence of processes that violate the combined charge conjugation and parity (CP) symmetry[1]. As such, precision tests of CP symmetry may be used to search for physics beyond the standard model. However, hadrons decay through an interplay of strong and weak processes, quantified in terms of relative phases between the amplitudes. Although previous experiments constructed CP observables that depend on both strong and weak phases, we present an approach where sequential two-body decays of entangled multi-strange baryon–antibaryon pairs provide a separation between these phases. Our method, exploiting spin entanglement between the double-strange $\Xi^-$ baryon and its antiparticle[2] $\overline{\Xi}^+$, has enabled a direct determination of the weak-phase difference, $(\xi_P - \xi_S) = (1.2 \pm 3.4 \pm 0.8) \times 10^{-2}$ rad. Furthermore, three independent CP observables can be constructed from our measured parameters. The precision in the estimated parameters for a given data sample size is several orders of magnitude greater than achieved with previous methods[3]. Finally, we provide an independent measurement of the recently debated $\Lambda$ decay parameter $\alpha_\Lambda$ (refs. [4,5]). The $\Lambda\overline{\Lambda}$ asymmetry is in agreement with and compatible in precision to the most precise previous measurement[4].

Small violations of CP symmetry are predicted by the standard model[6,7] and are a well established phenomenon in weak decays of mesons. However, the mechanisms of the standard model are too specific to yield effects of a size that can explain the observed matter–antimatter asymmetry of the Universe[8,9]. Therefore, CP tests can be considered a promising area to search for physics beyond the standard model[10,11]. So far, no CP-violating effects beyond the standard model have been observed in the baryon sector[12].

In general, CP symmetry is tested by comparing the decay patterns of a particle to those of its antiparticle. Many CP-symmetry tests in hadron decays rely on strong interactions of the final particles to reveal the signal. This strategy is applied in the determination of the ratio $\varepsilon'/\varepsilon$, quantifying the difference between the two-pion decay rates of the two weak eigenstates of neutral kaons. The $\varepsilon'/\varepsilon$ measurement constitutes the only observation of direct CP violation for light strange hadrons[13,14] and provides the most stringent test of contributions beyond the standard model in strange quark systems[15]. This strategy, however, comes at a price: it is difficult to disentangle, in a model-independent way, the contributions from weak interactions or processes beyond the standard model from those of strong processes. Approaches that do not rely on strong interactions require that the kaon decay into four final-state particles[16].

Baryons provide additional information through spin measurements. Known examples involving three-body decays are spin correlations and polarization in nuclear and neutron $\beta$ decays[17]. Sequential two-body decays of entangled multi-strange baryon–antibaryon pairs provide

another, hitherto unexplored, diagnostic tool to separate the strong and the weak phases.

In this work we explore spin correlations in weak two-body decays of spin-½ baryons. The spin direction of the parent baryon manifests itself in the momentum direction of the daughter particle, enabling straightforward experimental access to the spin properties. Spin-½ baryon decays are described by a parity-conserving (P-wave) and a parity-violating (S-wave) amplitude, quantified in terms of the decay parameters $\alpha_Y$, $\beta_Y$ and $\gamma_Y$ (ref. [18]). The $Y$ refers to the decaying parent hyperon (for example, $\Lambda$ or $\Xi$). These parameters are constrained by the relation $\alpha_Y^2 + \beta_Y^2 + \gamma_Y^2 = 1$. By defining the parameter $\phi_Y$ according to

$$\beta_Y = \sqrt{1-\alpha_Y^2}\,\sin\phi_Y, \quad \gamma_Y = \sqrt{1-\alpha_Y^2}\,\cos\phi_Y, \qquad (1)$$

the decay is completely described by two independent parameters $\alpha_Y$ and $\phi_Y$. In the standard experimental approach[3,4,19–21], the initial baryon is produced in a well defined spin polarized state, which allows access to the decay parameters through the angular distribution of the final-state particles. For sequentially decaying baryons, for example, the decay of the double-strange $\Xi^-$ baryon into $\Lambda\pi^-$, two effects are possible: 1) a polarized $\Xi^-$ transfers its polarization $\mathbf{P}_\Xi$ to the daughter $\Lambda$; 2) a longitudinal component of the daughter $\Lambda$ polarization is induced by the $\Xi^-$ decay, even if the $\Xi^-$ polarization has no component in this direction. In a reference system with the $\hat{\mathbf{z}}$ axis along the $\Lambda$ momentum in the $\Xi^-$ rest frame and the $\hat{\mathbf{y}}$ axis along $\mathbf{P}_\Xi \times \hat{\mathbf{z}}$, the $\Lambda$ polarization vector is given by[18]

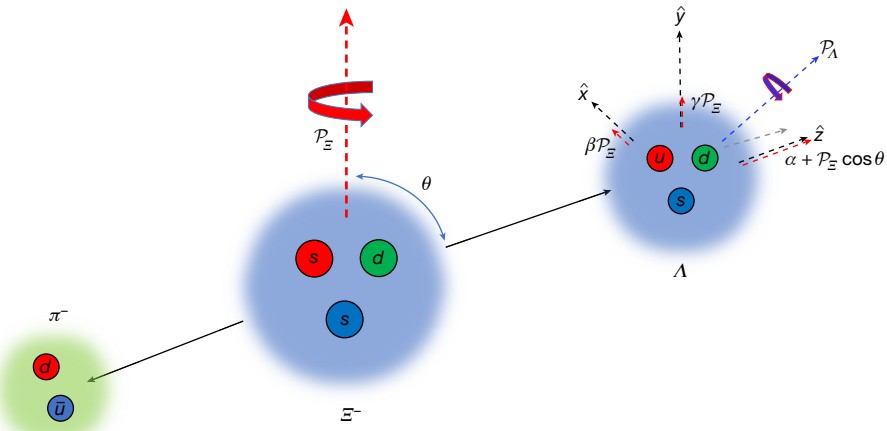

**Fig. 1 | Illustration of the polarization vectors of $\Xi^-$ and $\Lambda$ in relation to the decay parameters $\alpha$, $\beta$ and $\gamma$ of the $\Xi^- \to \Lambda\pi^-$ decay.** The $\Lambda$ polarization $\mathcal{P}_\Lambda$ has a component in the longitudinal as well as the transverse direction, where the former ($\hat{z}$) is defined by the $\Lambda$ momentum. The longitudinal component

depends on the $\Lambda$ emission angle and arises from the transferred $\Xi^-$ polarization $\mathcal{P}_\Xi$ combined with the decay parameter $\alpha$. The remaining $\Xi^-$ polarization is transferred to the transverse components according to $\beta\mathcal{P}_\Xi$ ($\hat{x}$) and $\gamma_\Xi\mathcal{P}_\Xi$ ($\hat{y}$). Quarks: $d$, down; $s$, strange; $u$, up; $\bar{u}$, antiup.

as illustrated in Fig. 1. This means that the longitudinal ($\hat{z}$) component depends on $\alpha_\Xi$, and the transversal components are rotated by the angle $\phi_\Xi$ with respect to the $\Xi^-$ polarization.

The decay parameter $\alpha_\Xi$ appears explicitly in the angular distribution of the direct decay $\Xi^- \to \Lambda\pi^-$, whereas the sequential decay distribution of the daughter $\Lambda$ depends on both $\alpha_\Lambda$ and $\phi_\Xi$. CP symmetry implies that the baryon decay parameters $\alpha$ and $\phi$ equal those of the antibaryon $\bar{\alpha}$ and $\bar{\phi}$ but with opposite sign. Hence, CP violation can be quantified in terms of the observables

$$A_{CP}^Y = \frac{\alpha_Y + \bar{\alpha}_Y}{\alpha_Y - \bar{\alpha}_Y}, \quad \Delta\phi_{CP} = \frac{\phi_Y + \bar{\phi}_Y}{2}. \tag{3}$$

CP violation can only be observed if there is interference between CP-even and CP-odd terms in the decay amplitude. Because the decay amplitude for $\Xi^- \to \Lambda\pi^-$ consists of both a P-wave and an S-wave part, the leading-order contribution to the CP asymmetry, $A_{CP}^\Xi$, can be written as

$$A_{CP}^\Xi \approx -\tan(\delta_P - \delta_S)\tan(\xi_P - \xi_S), \tag{4}$$

where $\tan(\delta_P - \delta_S) = \beta/\alpha$ denotes the strong-phase difference of the final-state interaction between the $\Lambda$ and $\pi^-$ from the $\Xi^-$ decay. CP-violating effects would manifest themselves in a nonzero weak-phase difference $\xi_P - \xi_S$ (refs. [22–24]), an observable that is complementary to the kaon decay parameter $\varepsilon'$ (refs. [13,14,25]) because the latter only involves an S-wave. The strong-phase difference can be extracted from the $\phi_\Xi$ parameter, and is found to be small[3,26]: $(-0.037 \pm 0.014)$. Hence, CP-violating signals in $A_{CP}^\Xi$ are strongly suppressed and difficult to interpret in terms of the weak-phase difference.

An independent CP-symmetry test in $\Xi^- \to \Lambda\pi^-$ is provided by determining the value of $\Delta\phi_{CP}$. At leading order, this observable is related directly to the weak-phase difference:

$$(\xi_P - \xi_S)_{LO} = \frac{\beta + \bar{\beta}}{\alpha - \bar{\alpha}} \approx \frac{\sqrt{1 - \langle\alpha\rangle^2}}{\langle\alpha\rangle}\Delta\phi_{CP}, \tag{5}$$

where $\langle\alpha\rangle = (\alpha - \bar{\alpha})/2$, and can be measured even if $\delta_P = \delta_S$. The absence of a strong suppression factor therefore improves the sensitivity to CP-violation effects by an order of magnitude with respect to that of the $A_{CP}^\Xi$ observable[22,23]. To measure $\Delta\phi_{CP}$ using the standard polarimeter technique from refs. [21,28] requires beams of polarized $\Xi^-$ and $\bar{\Xi}^+$. In such experiments the precision is limited by the magnitude of the polarization and the accuracy of the polarization determination, which in turn is sensitive to asymmetries in the production mechanisms[27]. In fact, no experiment with a polarized $\bar{\Xi}^+$ has been performed, and the polarization of the $\Xi^-$ beams were below 5% (ref. [3]). Here we present an alternative approach, in which the baryon–antibaryon pair is produced in a spin-entangled CP eigenstate and all decay sequences are analysed simultaneously.

To the best of our knowledge, no direct measurements of any of the asymmetries defined in equation (3) have been performed for the $\Xi^-$ baryon. The HyperCP experiment[28], designed for the purpose of CP tests in baryon decays, used samples of around $10^7$–$10^8$ $\Xi^-$ and $\bar{\Xi}^+$ events to determine the products $\alpha_\Xi\alpha_\Lambda$ and $\bar{\alpha}_\Xi\bar{\alpha}_\Lambda$. From these measurements, the sum $A_{CP}^\Lambda + A_{CP}^\Xi$ was estimated to be $(0.0 \pm 5.5 \pm 4.4) \times 10^{-4}$, where the first uncertainty is statistical and the second systematic. In addition to the aforementioned problem of the smallness of $\phi_\Xi$, which limits the sensitivity of $A_{CP}^\Xi$ to CP violation, an observable defined as the sum of asymmetries comes with other drawbacks: if $A_{CP}^\Lambda$ and $A_{CP}^\Xi$ have opposite signs, the sum could be consistent with zero even in the presence of CP-violating effects. A precise interpretation therefore requires an independent measurement of $A_{CP}^\Lambda$ with matching precision. The most precise result so far is a recent BESIII measurement[4] where $A_{CP}^\Lambda$ was found to be $(-6 \pm 12 \pm 7) \times 10^{-3}$. Furthermore, ref. [4] revealed a 17% disagreement with previous measurements on the $\alpha_\Lambda$ parameter[26], a result that rapidly gained some support from a re-analysis of CLAS data[5]. Although the CLAS result is in better agreement with BESIII than with the Particle Data Group value from 2018 and earlier, there is a discrepancy between the CLAS and BESIII results that needs to be understood. This is particularly important because many physics quantities from various fields depend on the parameter $\alpha_\Lambda$. Examples include baryon spectroscopy, heavy-ion physics and hyperon-related studies at the Large Hadron Collider[29–34].

In this work we apply a newly designed method[2,35] to study entangled, sequentially decaying baryon–antibaryon pairs in the process $e^+e^- \to J/\psi \to \Xi^-\bar{\Xi}^+$. This approach enables a direct measurement of all weak decay parameters of the $\Xi^- \to \Lambda\pi^-$, $\Lambda \to p\pi^-$ decay, and the corresponding parameters of the $\bar{\Xi}^+$. The production and multi-step decays can be described by nine kinematic variables, here expressed as the helicity angles $\boldsymbol{\xi} = (\theta, \theta_{\bar{\Lambda}}, \varphi_\Lambda, \theta_{\bar{\Lambda}}, \varphi_{\bar{\Lambda}}, \theta_p, \varphi_p, \theta_{\bar{p}}, \varphi_{\bar{p}})$. The first, $\theta$, is the

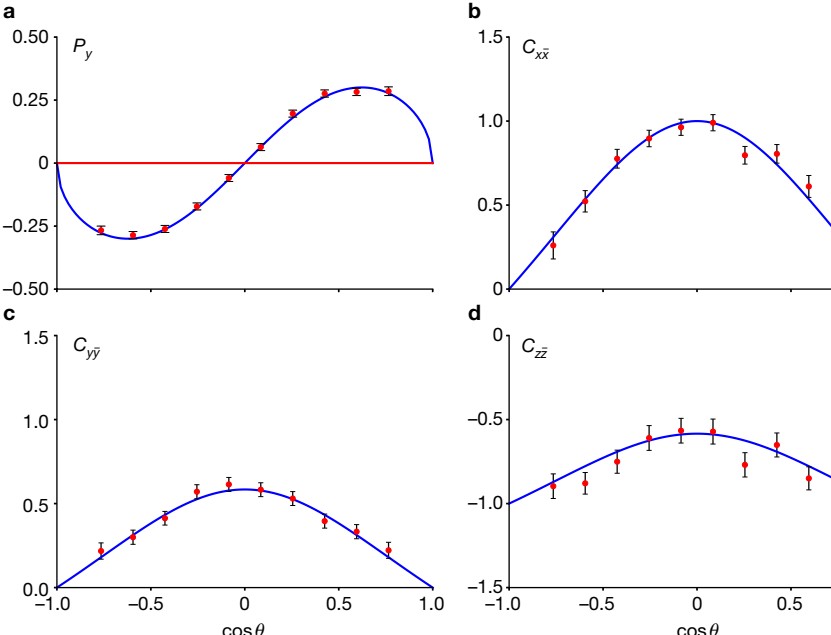

**Fig. 2 | Polarization in and spin correlations of the $e^+e^- \to \Xi^- \overline{\Xi}^+$ reaction.** **a**, Polarization in the $e^+e^- \to \Xi^- \overline{\Xi}^+$ reaction. **b**–**d**, Spin correlations of the $e^+e^- \to \Xi^- \overline{\Xi}^+$ reaction. The coordinate systems $\mathcal{R}_\Xi$ and $\mathcal{R}_{\overline{\Xi}}$ of the $\Xi^-$ and $\overline{\Xi}^+$, respectively, are described in the text. The data points are determined

independently in each bin of the $\Xi^-$ cosine scattering angle in the $e^+e^-$ centre-of-momentum system. The blue curves represent the expected angular dependence obtained with the production parameters $\alpha_\psi$ and $\Delta\Phi$ from the global maximum log-likelihood fit. The error bars indicate the statistical uncertainties.

$\Xi^-$ scattering angle with respect to the $e^+$ beam in the centre-of-momentum system of the reaction. The angles $\theta_\Lambda$ and $\varphi_\Lambda$ ($\theta_{\overline{\Lambda}}$, $\varphi_{\overline{\Lambda}}$) are defined by the $\Lambda$ ($\overline{\Lambda}$) direction in a reference system denoted $\mathcal{R}_\Xi$ ($\mathcal{R}_{\overline{\Xi}}$), where $\Xi^-$ ($\overline{\Xi}^+$) is at rest and where the $\hat{\mathbf{z}}$ axis points in the direction of the $\Xi^-$ ($\overline{\Xi}^+$) in the centre-of-momentum system. The $\hat{\mathbf{y}}$ axis is normal to the production plane. The angles $\theta_p$ and $\varphi_p$ ($\theta_{\overline{p}}$ and $\varphi_{\overline{p}}$) give the direction of the proton (antiproton) in the $\Lambda$ ($\overline{\Lambda}$) rest system, denoted $\mathcal{R}_\Lambda$ ($\mathcal{R}_{\overline{\Lambda}}$), with the $\hat{\mathbf{z}}$ axis pointing in the direction of the $\Lambda$ ($\overline{\Lambda}$) in the $\mathcal{R}_\Xi$ ($\mathcal{R}_{\overline{\Xi}}$) system and the $\hat{\mathbf{y}}$ axis normal to the plane spanned by the direction of the $\Xi^-$ ($\overline{\Xi}^+$) and the direction of the $\Lambda$ ($\overline{\Lambda}$). The structure of the nine-dimensional angular distribution is determined by eight global (that is, independent of the $\Xi^-$ scattering angle) parameters $\boldsymbol{\omega} = (\alpha_\psi, \Delta\Phi, \alpha_\Xi, \phi_\Xi, \bar{\alpha}_\Xi, \bar{\phi}_\Xi, \alpha_\Lambda, \bar{\alpha}_\Lambda)$, and can be written in a modular form as[35]:

$$\mathcal{W}(\boldsymbol{\xi}; \boldsymbol{\omega}) = \sum_{\mu,\nu=0}^{3} C_{\mu\nu} \sum_{\mu'\nu'=0}^{3} a_{\mu\mu}^{\Xi} a_{\nu\nu}^{\overline{\Xi}} a_{\mu',0}^{\Lambda} a_{\nu',0}^{\overline{\Lambda}}. \tag{6}$$

Here $C_{\mu\nu}(\theta; \alpha_\psi, \Delta\Phi)$ is a $4 \times 4$ spin density matrix, defined in the aforementioned reference systems $\mathcal{R}_\Xi$ and $\mathcal{R}_{\overline{\Xi}}$, describing the spin configuration of the entangled hyperon–antihyperon pair. The parameters $\alpha_\psi$ and $\Delta\Phi$ are related to two production amplitudes, where $\alpha_\psi$ parameterizes the $\Xi^-$ angular distribution. The $\Delta\Phi$ is the relative phase between the two production amplitudes (in the so-called helicity representation)[36] and governs the polarization $P_y$ of the produced $\Xi^-$ and $\overline{\Xi}^+$ as well as their spin correlations $C_{ij}$. The matrix elements are related to $P_y = P_y(\theta)$ and $C_{ij} = C_{ij}(\theta)$ in the following way:

$$C_{\mu\nu} = (1 + \alpha_\psi \cos^2\theta) \begin{pmatrix} 1 & 0 & P_y & 0 \\ 0 & C_{xx} & 0 & C_{xz} \\ -P_y & 0 & C_{yy} & 0 \\ 0 & -C_{xz} & 0 & C_{zz} \end{pmatrix}. \tag{7}$$

The matrices $a_{\mu\nu}^Y$ in equation (6) represent the propagation of the spin density matrices in the sequential decays. The elements of these

4 × 4 matrices are parameterized in terms of the weak decay parameters $\alpha_Y$ and $\phi_Y$ as well as the helicity angles: $a_{\mu\mu}^{\overline{\Xi}}(\theta_\Lambda, \varphi_\Lambda; \alpha_\Xi, \phi_\Xi)$ in reference system $\mathcal{R}_\Xi$, $a_{\nu\nu}^{\overline{\Xi}}(\theta_{\overline{\Lambda}}, \varphi_{\overline{\Lambda}}; \bar{\alpha}_\Xi, \bar{\phi}_\Xi)$ in system $\mathcal{R}_{\overline{\Xi}}$, $a_{\mu,0}^{\Lambda}(\theta_p, \varphi_p; \bar{\alpha}_\Lambda)$ in system $\mathcal{R}_\Lambda$, and $a_{\nu,0}^{\Lambda}(\theta_{\overline{p}}, \varphi_{\overline{p}}; \bar{\alpha}_\Lambda)$ in system $\mathcal{R}_{\overline{\Lambda}}$. The full expressions of $C_{\mu\nu}$ and $a_{\mu\nu}^Y$ are given in ref.[35].

We have carried out our analysis on a data sample of $(1.3106 \pm 0.0070) \times 10^9 J/\psi$ events collected in electron–positron annihilations with the multi-purpose BESIII detector[37]. The $J/\psi$ resonance decays into the $\Xi^- \overline{\Xi}^+$ final state with a branching fraction[26] of $(9.7 \pm 0.8) \times 10^{-4}$. Our method requires exclusively reconstructed $\Xi^- \overline{\Xi}^+ \to \Lambda\pi^- \overline{\Lambda}\pi^+ \to p\pi^-\pi^- \overline{p}\pi^+\pi^+$

**Table 1 | Summary of results**

| Parameter | This work | Previous result | Reference |
|---|---|---|---|
| $\alpha_\psi$ | 0.586±0.012±0.010 | 0.58±0.04±0.08 | Ref.[49] |
| $\Delta\Phi$ | 1.213±0.046±0.016 rad | – | |
| $\alpha_\Xi$ | −0.376±0.007±0.003 | −0.401±0.010 | Ref.[26] |
| $\phi_\Xi$ | 0.011±0.019±0.009 rad | −0.037±0.014 rad | Ref.[26] |
| $\bar{\alpha}_\Xi$ | 0.371±0.007±0.002 | – | |
| $\bar{\phi}_\Xi$ | −0.021±0.019±0.007 rad | – | |
| $\alpha_\Lambda$ | 0.757±0.011±0.008 | 0.750±0.009±0.004 | Ref.[4] |
| $\overline{\alpha}_\Lambda$ | −0.763±0.011±0.007 | −0.758±0.010±0.007 | Ref.[4] |
| $\xi_P - \xi_S$ | (1.2±3.4±0.8)×10⁻² rad | – | |
| $\delta_P - \delta_S$ | (−4.0±3.3±1.7)×10⁻² rad | (10.2±3.9)×10⁻² rad | Ref.[3] |
| $A_{CP}^\Xi$ | (6±13±6)×10⁻³ | – | |
| $\Delta\phi_{CP}^\Xi$ | (−5±14±3)×10⁻³ rad | – | |
| $A_{CP}^\Lambda$ | (−4±12±9)×10⁻³ | (−6±12±7)×10⁻³ | Ref.[4] |
| $\langle\phi_\Xi\rangle$ | 0.016±0.014±0.007 rad | | |

The $J/\psi \to \Xi^- \overline{\Xi}^+$ angular distribution parameter $\alpha_\psi$, the hadronic form factor phase $\Delta\Phi$, the decay parameters for $\Xi^- \to \Lambda\pi^-$ ($\alpha_\Xi, \phi_\Xi$), $\overline{\Xi}^+ \to \overline{\Lambda}\pi^+$ ($\bar{\alpha}_\Xi, \bar{\phi}_\Xi$) $\Lambda \to p\pi^-$ ($\alpha_\Lambda$) and $\overline{\Lambda} \to \overline{p}\pi^+$ ($\bar{\alpha}_\Lambda$); the CP asymmetries $A_{CP}^\Xi$, $\Delta\phi_{CP}^\Xi$ and $A_{CP}^\Lambda$, and the average $\langle\phi_\Xi\rangle$. The first and second uncertainties are statistical and systematic, respectively.

events. The final-state particles are measured in the main drift chamber, where a superconducting solenoid provides a magnetic field allowing momentum determination with an accuracy of 0.5% at 1.0 GeV/$c$. The $\Lambda$ ($\overline{\Lambda}$) candidates are identified by combining $p\pi^-$ ($\overline{p}\pi^+$) pairs and the $\Xi^-$ ($\overline{\Xi}^+$) candidates by subsequently combining $\Lambda\pi^-$ ($\overline{\Lambda}\pi^+$) pairs. Because it was found that the long-lived $\Xi^-$ and $\overline{\Xi}^+$ can only be reconstructed with sufficient quality if they fulfil $|\cos\theta| < 0.84$, only $\Xi^-$ and $\overline{\Xi}^+$ reconstructed within this range were considered. After applying all selection criteria, 73,244 $\Xi^-\overline{\Xi}^+$ event candidates remain in the sample. The number of background events in the signal is estimated to be 199 ± 17. More details of the analysis are given in Methods.

For each event, the complete set of the kinematic variables $\boldsymbol{\xi}$ is calculated from the intermediate and final-state particle momenta. The physical parameters in $\boldsymbol{\omega}$ are then determined from $\boldsymbol{\xi}$ by an unbinned maximum log-likelihood fit where the multidimensional reconstruction efficiency is taken into account. The details of the maximum log-likelihood fit procedure and the systematic uncertainties are described in Methods.

The results of the fit, that is, the weak decay parameters $\Xi^- \to \Lambda\pi^-$ and $\overline{\Xi}^+ \to \overline{\Lambda}\pi^+$, as well as the production-related parameters $\alpha_\psi$ and $\Delta\Phi$, are summarized in Table 1. To illustrate the fit quality, the diagonal spin correlations and the polarization defined in equation (7) are shown in Fig. 2. The upper-left panel of Fig. 2 shows that the $\Xi^-$ baryon is polarized with respect to the normal of the production plane. The maximum polarization is approximately 30%, as shown in the figure. The data points are determined by independent fits for each $\cos\theta$ bin, without any assumptions on the $\cos\theta$ dependence of $C_{\mu\nu}$. The red curves represent the angular dependence obtained with the parameters $\alpha_\psi$ and $\Delta\Phi$ determined from the global maximum log-likelihood fit. The independently determined data points agree well with the globally fitted curves.

The extracted values of $\alpha_\Xi, \bar{\alpha}_\Xi, \phi_\Xi, \bar{\phi}_\Xi, \alpha_\Lambda$ and $\bar{\alpha}_\Lambda$ and their correlations allow for three independent CP symmetry tests. The asymmetry $A_{\mathrm{CP}}^\Xi$ is measured for the first time and found to be $(6 \pm 13 \pm 6) \times 10^{-3}$, where the first uncertainty is statistical and the second systematic. The corresponding standard model (SM) prediction[24] is $A_{\mathrm{CP,SM}}^\Xi = (-0.6 \pm 1.6) \times 10^{-5}$.

The result for the decay parameter $\bar{\phi}_\Xi$ is, to our knowledge, the first measurement of its kind for a weakly decaying antibaryon. By combining this parameter with the corresponding $\phi_\Xi$ measurement, the CP asymmetry $\Delta\phi_{\mathrm{CP}}^\Xi$ can be determined, and is found to be $(-5 \pm 14 \pm 3) \times 10^{-3}$ rad. Because this result is consistent with zero, we can improve our knowledge of the value of $\phi_\Xi$ by assuming CP symmetry and then calculating the mean value of $\phi_\Xi$ and $\bar{\phi}_\Xi$. This procedure yields $\langle\phi_\Xi\rangle = 0.016 \pm 0.014 \pm 0.007$ rad, which differs from the HyperCP measurement, $\phi_{\Xi,\mathrm{HyperCP}} = -0.042 \pm 0.011 \pm 0.011$ rad, by 2.6 standard deviations[3]. It is noteworthy that our method yields a precision in $\langle\phi_\Xi\rangle$ that is similar to that of the HyperCP result, despite the three-orders-of-magnitude larger data sample of the latter measurement. This demonstrates the intrinsically high sensitivity that can be achieved with entangled baryon–antibaryon pairs.

The measurement of $\langle\phi_\Xi\rangle$, together with the mean value $\langle\alpha_\Xi\rangle = -0.373 \pm 0.005 \pm 0.002$, enables a direct determination of the strong-phase difference, which is found to be $(\delta_\mathrm{P} - \delta_\mathrm{S}) = (-4.0 \pm 3.3 \pm 1.7) \times 10^{-2}$ rad. This is consistent with the standard model predictions obtained in the framework of heavy-baryon chiral perturbation theory[24] of $(1.9 \pm 4.9) \times 10^{-2}$ rad but in disagreement with the value $(10.2 \pm 3.9) \times 10^{-2}$ rad that one obtains from the HyperCP $\phi_\Xi$ measurement[3] by using $\alpha_\Xi = -0.376$ from this work. Because the $(\delta_\mathrm{P} - \delta_\mathrm{S})$ value obtained in our analysis is consistent with zero, a calculation of the weak-phase difference from equation (4) is unfeasible. Instead, we apply equation (5), which yields $(\xi_\mathrm{P} - \xi_\mathrm{S}) = (1.2 \pm 3.4 \pm 0.8) \times 10^{-2}$ rad. This is one of the most precise tests of the CP symmetry for strange baryons and the first direct measurement of the weak-phase difference for any baryon. The corresponding standard model prediction[24] is $(\xi_\mathrm{P} - \xi_\mathrm{S})_\mathrm{SM} = (1.8 \pm 1.5) \times 10^{-4}$ rad.

Sequential $\Xi^-$ decays also provide an independent measurement of the $\Lambda$ decay parameters $\alpha_\Lambda$ and $\bar{\alpha}_\Lambda$. Being the lightest baryon with strangeness, $\Lambda$ appear in the decay chain of many other baryons ($\Sigma^0, \Xi^0, \Xi^-, \Omega, \Lambda_\mathrm{c}$ and so on) decay with appreciable fractions into final states containing $\Lambda$. The measurements of spin observables[38–40] and decay parameters of heavier baryons[41,42] therefore implicitly depend on $\alpha_\Lambda$. Furthermore, because decaying $\Lambda$ and $\overline{\Lambda}$ beams are used for producing polarized proton and antiproton beams[43], all physics from such experiments rely on a correct determination of $\alpha_\Lambda$. The value of $\alpha_\Lambda = 0.757 \pm 0.011 \pm 0.008$ measured in this analysis is in excellent agreement with that obtained from the $J/\psi \to \Lambda\overline{\Lambda}$ analysis of BESIII[4], although it disagrees with the result from the re-analysis of CLAS data[5]. The precision of our measurement is similar to that of the $J/\psi \to \Lambda\overline{\Lambda}$ study[4], despite being based on a data sample that was six times smaller. The larger sensitivity is primarily explained by the fact that $\alpha_\Lambda$ in equation (6) appears in a product with the polarization, which is much larger in the case of $\Lambda$ baryons from $\Xi^-$ decays compared to those directly produced in $J/\psi \to \Lambda\overline{\Lambda}$. Furthermore, the multi-step process enhances the angular correlations between the baryons and antibaryons to such an extent that $\alpha_\Xi$ and $\alpha_\Lambda$ can be measured with the same precision even if the $\Xi^-\overline{\Xi}^+$ pair is produced unpolarized[2].

To summarize, this Article presents a very sensitive test of CP symmetry. This test provides a hunting ground for physics beyond the standard model in strange hadrons that is complementary to $\varepsilon'/\varepsilon$ measurements in kaon decays[44]. The contributions to $\varepsilon$ and $\varepsilon'$, from hyperon decays on the one hand and kaon decays on the other, are described by different combinations of quark operators. In addition, hyperons provide information on the spin structure of the operators that is not possible to obtain from kaon decays. When applied to future measurements with larger datasets at BESIII[45], the upcoming PANDA experiment at FAIR[46] and the proposed Super-Charm Tau Factory projects in China and Russia[47,48], our method has potential to reach the required precision for CP-violating signals, provided such effects exist.

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

**The BESIII Collaboration**

M. Ablikim[1], M. N. Achasov[2,78], P. Adlarson[3], S. Ahmed[4], M. Albrecht[5], R. Aliberti[6], A. Amoroso[7,8], M. R. An[9], Q. An[10,11], X. H. Bai[12], Y. Bai[13], O. Bakina[14], R. Baldini Ferroli[15], I. Balossino[16], Y. Ban[17,79], K. Begzsuren[18], N. Berger[6], M. Bertani[15], D. Bettoni[16], F. Bianchi[7,8], J. Biernat[3], J. Bloms[19], A. Bortone[7,8], I. Boyko[14], R. A. Briere[20], H. Cai[21], X. Cai[1,11], A. Calcaterra[15], G. F. Cao[1,22], N. Cao[1,22], S. A. Cetin[23], J. F. Chang[1,11], W. L. Chang[1,22], G. Chelkov[14,80], D. Y. Chen[24], G. Chen[1], H. S. Chen[1,22], M. L. Chen[1,11], S. J. Chen[25], X. R. Chen[26], Y. B. Chen[1,11], Z. J. Chen[27,81], W. S. Cheng[8], G. Cibinetto[16], F. Cossio[7,8], X. F. Cui[28], H. L. Dai[1,11], X. C. Dai[1,22], A. Dbeyssi[4], R. E. de Boer[5], D. Dedovich[14], Z. Y. Deng[1], A. Denig[6], I. Denysenko[14], M. Destefanis[7,8], F. De Mori[7,8], Y. Ding[29], C. Dong[28], J. Dong[1,11], L. Y. Dong[1,22], M. Y. Dong[1,11,22], X. Dong[21], S. X. Du[30], Y. L. Fan[21], J. Fang[1,11], S. S. Fang[1,22], Y. Fang[1], R. Farinelli[16], L. Fava[8,31], F. Feldbauer[5], G. Felici[15], C. Q. Feng[10,11], M. Fritsch[5], C. D. Fu[1], Y. Gao[17,79], Y. Gao[33], Y. Gao[10,11], Y. G. Gao[24], I. Garzia[16,34], P. T. Ge[21], C. Geng[32], E. M. Gersabeck[35], A. Gilman[36], K. Goetzen[37], L. Gong[29], W. X. Gong[1,11], W. Gradl[6], M. Greco[7,8], L. M. Gu[25], M. H. Gu[1,11], S. Gu[38], Y. T. Gu[39], C. Y. Guan[1,22], A. Q. Guo[40], L. B. Guo[41], R. P. Guo[42], Y. P. Guo[43,82,83], A. Guskov[14], T. T. Han[44], W. Y. Han[9], J. Hansson[3], X. Q. Hao[45], F. A. Harris[46], N. Hüsken[6,40], K. L. He[1,22], F. H. Heinsius[5], C. H. Heinz[6], T. Held[5], Y. K. Heng[1,11,22], C. Herold[47], M. Himmelreich[37,84], T. Holtmann[5], G. S. Huang[10,11], L. Q. Huang[33], X. T. Huang[44], Y. P. Huang[1], Z. Huang[17,79], T. Hussain[49], W. Ikegami Andersson[3], W. Imoehl[40], M. Irshad[10,11], S. Jaeger[5], S. Janchiv[18,87], Q. Ji[1], Q. P. Ji[45], X. B. Ji[1,22], X. L. Ji[1,11], Y. Y. Ji[44], H. B. Jiang[44], X. S. Jiang[1,11,22], J. B. Jiao[44], Z. Jiao[50], S. Jin[25], Y. Jin[12], T. Johansson[3], N. Kalantar-Nayestanaki[51], X. S. Kang[29], R. Kappert[51], M. Kavatsyuk[51], B. C. Ke[1,52], I. K. Keshk[5], A. Khoukaz[19], P. Kiese[6], R. Kiuchi[1], R. Kliemt[37], L. Koch[53], O. B. Kolcu[23], B. Kopf[5], M. Kuemmel[5], M. Kuessner[5], A. Kupsc[3], M. G. Kurth[1,22], W. Kühn[53], J. J. Lane[35], J. S. Lange[53], P. Larin[4], A. Lavania[54], L. Lavezzi[7,8], Z. H. Lei[10,11], H. Leithoff[6], M. Lellmann[6], T. Lenz[6], C. Li[55], C. H. Li[9], Cheng Li[10,11], D. M. Li[30], F. Li[1,11], G. Li[1], H. Li[10,11], H. Li[52], H. B. Li[1,22], H. J. Li[45], H. N. Li[43,82,83], J. L. Li[44], J. Q. Li[5], J. S. Li[32], Ke Li[1], L. K. Li[1], Lei Li[56], P. R. Li[57], S. Y. Li[58], W. D. Li[1,22], W. G. Li[1], X. H. Li[10,11], X. L. Li[44], Xiaoyu Li[1,22], Z. Y. Li[32], H. Liang[1,22], H. Liang[10,11], H. Liang[59], Y. F. Liang[60], Y. T. Liang[26], G. R. Liao[61], L. Z. Liao[1,22], J. Libby[54], C. X. Lin[32], B. J. Liu[1], C. X. Liu[1], D. Liu[10,11], F. H. Liu[62], Fang Liu[1], Feng Liu[24], H. B. Liu[39], H. M. Liu[1,22], Huanhuan Liu[1], Huihui Liu[63], J. B. Liu[10,11], J. L. Liu[33], J. Y. Liu[1,22], K. Liu[1], K. Y. Liu[29], Ke Liu[24], L. Liu[10,11], M. H. Liu[43,82,83], P. L. Liu[1], Q. Liu[22], S. B. Liu[10,11], Shuai Liu[64], T. Liu[1,22], W. M. Liu[10,11], X. Liu[57], Y. Liu[57], Y. B. Liu[28], Z. A. Liu[1,11,22], Z. Q. Liu[44], X. C. Lou[1,11,22], F. X. Lu[45], F. X. Lu[32], H. J. Lu[50], J. D. Lu[1,22], J. G. Lu[1,11], X. L. Lu[1], Y. Lu[1], Y. P. Lu[1,11], C. L. Luo[41], M. X. Luo[65], P. W. Luo[32], T. Luo[43,82,83], X. L. Luo[1,11], S. Lusso[8], X. R. Lyu[22], F. C. Ma[29], H. L. Ma[1], L. L. Ma[44], M. M. Ma[1,22], Q. M. Ma[1], R. Q. Ma[1,22], R. T. Ma[22], X. X. Ma[1,22], X. Y. Ma[1,11], F. E. Maas[4], M. Maggiora[7,8], S. Maldaner[5], S. Malde[36], Q. A. Malik[49], A. Mangoni[66], Y. J. Mao[17,79], Z. P. Mao[1], S. Marcello[7,8], Z. X. Meng[12], J. G. Messchendorp[51], G. Mezzadri[16], T. J. Min[25], R. E. Mitchell[40], X. H. Mo[1,11,22], Y. J. Mo[24], N. Yu. Muchnoi[2,78], H. Muramatsu[67], S. Nakhoul[37,84], Y. Nefedov[14], F. Nerling[37,84], I. B. Nikolaev[2,78], Z. Ning[1,11], S. Nisar[68,88], S. L. Olsen[22], Q. Ouyang[1,11,22], S. Pacetti[66,69], X. Pan[43,82,83], Y. Pan[35], A. Pathak[1], P. Patteri[15], M. Pelizaeus[5], H. P. Peng[10,11], K. Peters[37,84], J. L. Ping[41], R. G. Ping[1,22], R. Poling[67], V. Prasad[10,11], H. Qi[10,11], H. R. Qi[58], K. H. Qi[26], M. Qi[25], T. Y. Qi[43], T. Y. Qi[38], S. Qian[1,11], W. B. Qian[22], Z. Qian[32], C. F. Qiao[22], L. Q. Qin[61], X. P. Qin[43], X. S. Qin[44], Z. H. Qin[1,11], J. F. Qiu[1], S. Q. Qu[28], K. H. Rashid[49], K. Ravindran[54], C. F. Redmer[6], A. Rivetti[8], V. Rodin[51], M. Rolo[8], G. Rong[1,22], Ch. Rosner[4], M. Rump[19], H. S. Sang[10], A. Sarantsev[14,89], Y. Schelhaas[6], C. Schnier[5], K. Schönning[3], M. Scodeggio[16,34], D. C. Shan[64], W. Shan[70], X. Y. Shan[10,11], J. F. Shangguan[64], M. Shao[10,11], C. P. Shen[43], P. X. Shen[28], X. Y. Shen[1,22], H. C. Shi[10,11], R. S. Shi[1,22], X. Shi[1,11], X. D. Shi[10,11], J. J. Song[44], W. M. Song[1,59], Y. X. Song[17,79], S. Sosio[7,8], S. Spataro[7,8], K. X. Su[21], P. P. Su[64], F. F. Sui[44], G. X. Sun[1], H. K. Sun[1], J. F. Sun[45], L. Sun[21], S. S. Sun[1,22], T. Sun[1,22], W. Y. Sun[41], W. Y. Sun[59], X. Sun[27,81], Y. J. Sun[10,11], Y. K. Sun[10,11], Y. Z. Sun[1], Z. T. Sun[1], Y. H. Tan[21], Y. X. Tan[10,11], C. J. Tang[60], G. Y. Tang[1], J. Tang[32], J. X. Teng[10,11], V. Thoren[3], Y. T. Tian[26], I. Uman[71], B. Wang[1], C. W. Wang[25], D. Y. Wang[17,79], H. J. Wang[57], H. P. Wang[1,22], K. Wang[1,11], L. L. Wang[1], M. Wang[44], M. Z. Wang[17,79], Meng Wang[1,22], W. Wang[32], W. H. Wang[21], W. P. Wang[10,11], X. Wang[17,79], X. F. Wang[57], X. L. Wang[43,82,83], Y. Wang[32], Y. Wang[10,11], Y. D. Wang[72], Y. F. Wang[1,11,22], Y. Q. Wang[1], Y. Y. Wang[57], Z. Wang[1,11], Z. Y. Wang[1], Ziyi Wang[22], Zongyuan Wang[1,22], D. H. Wei[61], P. Weidenkaff[6], F. Weidner[19], S. P. Wen[1], D. J. White[35], U. Wiedner[5], G. Wilkinson[36], M. Wolke[3], L. Wollenberg[5], J. F. Wu[1,22], L. H. Wu[1], L. J. Wu[1,22], X. Wu[43,82,83], Z. Wu[1,11], L. Xia[10,11], H. Xiao[43,82,83], S. Y. Xiao[1], Z. J. Xiao[41], X. H. Xie[17,79], Y. G. Xie[1,11], Y. H. Xie[24], T. Y. Xing[1,22], G. F. Xu[1], Q. J. Xu[73], W. Xu[1,22], X. P. Xu[64], Y. C. Xu[22], F. Yan[43,82,83], W. B. Yan[10,11], W. C. Yan[30], Xu Yan[64], H. J. Yang[74,90,91,92], H. X. Yang[1], L. Yang[52], S. L. Yang[22], Y. X. Yang[61], Yifan Yang[1,22], Zhi Yang[26], M. Ye[1,11], M. H. Ye[75], J. H. Yin[1], Z. Y. You[32], B. X. Yu[1,11,22], C. X. Yu[28], G. Yu[1,22], J. S. Yu[27,81], T. Yu[33], C. Z. Yuan[1,22], L. Yuan[38], X. Q. Yuan[17,79], Y. Yuan[1], Z. Y. Yuan[32], C. X. Yue[9], A. A. Zafar[49], Y. Zeng[27,81], B. X. Zhang[1], Guangyi Zhang[45], H. Zhang[10], H. H. Zhang[32], H. H. Zhang[59], H. Y. Zhang[1,11], J. J. Zhang[52], J. L. Zhang[76], J. Q. Zhang[41], J. W. Zhang[1,11,22], J. Y. Zhang[1], J. Z. Zhang[1,22], Jianyu Zhang[1,22], Jiawei Zhang[1,22], L. M. Zhang[58], L. Q. Zhang[32], Lei Zhang[25], S. Zhang[32], S. F. Zhang[25], Shulei Zhang[27,81], X. D. Zhang[72], X. Y. Zhang[44], Y. Zhang[36], Y. H. Zhang[1,11], Y. T. Zhang[10,11], Yan Zhang[10,11], Yao Zhang[1], Yi Zhang[43,82,83], Z. H. Zhang[24], Z. Y. Zhang[21], G. Zhao[1], J. Zhao[9], J. Y. Zhao[1,22], J. Z. Zhao[1,11], Lei Zhao[10,11], Ling Zhao[1], M. G. Zhao[28], Q. Zhao[1], S. J. Zhao[30], Y. B. Zhao[1,11], Y. X. Zhao[26], Z. G. Zhao[10,11], A. Zhemchugov[14,80], B. Zheng[33], J. P. Zheng[1,11], Y. Zheng[17,79], Y. H. Zheng[22], B. Zhong[41], C. Zhong[33], L. P. Zhou[1,22], Q. Zhou[1,22], X. Zhou[21], X. K. Zhou[22], X. R. Zhou[10,11], X. Y. Zhou[9], A. N. Zhu[1,22], J. Zhu[28], K. Zhu[1], K. J. Zhu[1,11,22], S. H. Zhu[77], T. J. Zhu[76], W. J. Zhu[28], W. J. Zhu[43,82,83], Y. C. Zhu[10,11], Z. A. Zhu[1,22], B. S. Zou[1] & J. H. Zou[1]

[1]Institute of High Energy Physics, Beijing, People's Republic of China. [2]G.I. Budker Institute of Nuclear Physics SB RAS (BINP), Novosibirsk, Russia. [3]Uppsala University, Uppsala, Sweden. [4]Helmholtz Institute Mainz, Mainz, Germany. [5]Ruhr-University Bochum, Bochum, Germany. [6]Johannes Gutenberg University of Mainz, Mainz, Germany. [7]University of Turin, Turin, Italy. [8]INFN, Turin, Italy. [9]Liaoning Normal University, Dalian, People's Republic of China. [10]University of Science and Technology of China, Hefei, People's Republic of China. [11]State Key Laboratory of Particle Detection and Electronics, Hefei, People's Republic of China. [12]University of Jinan, Jinan, People's Republic of China. [13]Southeast University, Nanjing, People's Republic of China. [14]Joint Institute for Nuclear Research, Dubna, Russia. [15]INFN Laboratori Nazionali di Frascati, Frascati, Italy. [16]INFN Sezione di Ferrara, Ferrara, Italy. [17]Peking University, Beijing, People's

Republic of China. [18]Institute of Physics and Technology, Ulaanbaatar, Mongolia. [19]University of Muenster, Muenster, Germany. [20]Carnegie Mellon University, Pittsburgh, PA, USA. [21]Wuhan University, Wuhan, People's Republic of China. [22]University of Chinese Academy of Sciences, Beijing, People's Republic of China. [23]Istinye University, Sariyer, Turkey. [24]Central China Normal University, Wuhan, People's Republic of China. [25]Nanjing University, Nanjing, People's Republic of China. [26]Institute of Modern Physics, Lanzhou, People's Republic of China. [27]Hunan University, Changsha, People's Republic of China. [28]Nankai University, Tianjin, People's Republic of China. [29]Liaoning University, Shenyang, People's Republic of China. [30]Zhengzhou University, Zhengzhou, People's Republic of China. [31]University of Eastern Piedmont, Alessandria, Italy. [32]Sun Yat-Sen University, Guangzhou, People's Republic of China. [33]University of South China, Hengyang, People's Republic of China. [34]University of Ferrara, Ferrara, Italy. [35]University of Manchester, Manchester, UK. [36]University of Oxford, Oxford, UK. [37]GSI Helmholtzcentre for Heavy Ion Research GmbH, Darmstadt, Germany. [38]Beihang University, Beijing, People's Republic of China. [39]Guangxi University, Nanning, People's Republic of China. [40]Indiana University, Bloomington, IN, USA. [41]Nanjing Normal University, Nanjing, People's Republic of China. [42]Shandong Normal University, Jinan, People's Republic of China. [43]Fudan University, Shanghai, People's Republic of China. [44]Shandong University, Jinan, People's Republic of China. [45]Henan Normal University, Xinxiang, People's Republic of China. [46]University of Hawaii, Honolulu, HI, USA. [47]Suranaree University of Technology, Nakhon Ratchasima, Thailand. [48]South China Normal University, Guangzhou, People's Republic of China. [49]University of the Punjab, Lahore, Pakistan. [50]Huangshan College, Huangshan, People's Republic of China. [51]University of Groningen, Groningen, The Netherlands. [52]Shanxi Normal University, Linfen, People's Republic of China. [53]II. Physikalisches Institut, Justus-Liebig-Universität Giessen, Giessen, Germany. [54]Indian Institute of Technology Madras, Chennai, India. [55]Qufu Normal University, Qufu, People's Republic of China. [56]Beijing Institute of Petrochemical Technology, Beijing, People's Republic of China. [57]Lanzhou University, Lanzhou, People's Republic of China. [58]Tsinghua University, Beijing, People's Republic of China. [59]Jilin University, Changchun, People's Republic of China. [60]Sichuan University, Chengdu, People's Republic of China. [61]Guangxi Normal University, Guilin, People's Republic of China. [62]Shanxi University, Taiyuan, People's Republic of China. [63]Henan University of Science and Technology, Luoyang, People's Republic of China. [64]Soochow University, Suzhou, People's Republic of China. [65]Zhejiang University, Hangzhou, People's Republic of China. [66]INFN Sezione di Perugia, Perugia, Italy. [67]University of Minnesota, Minneapolis, MN, USA. [68]COMSATS University Islamabad, Lahore, Pakistan. [69]University of Perugia, Perugia, Italy. [70]Hunan Normal University, Changsha, People's Republic of China. [71]Near East University, North Cyprus, Mersin, Turkey. [72]North China Electric Power University, Beijing, People's Republic of China. [73]Hangzhou Normal University, Hangzhou, People's Republic of China. [74]Shanghai Jiao Tong University, Shanghai, People's Republic of China. [75]China Center of Advanced Science and Technology, Beijing, People's Republic of China. [76]Xinyang Normal University, Xinyang, People's Republic of China. [77]University of Science and Technology Liaoning, Anshan, People's Republic of China. [78]Present address: Novosibirsk State University, Novosibirsk, Russia. [79]Present address: State Key Laboratory of Nuclear Physics and Technology, Peking University, Beijing, People's Republic of China. [80]Present address: Moscow Institute of Physics and Technology, Moscow, Russia. [81]Present address: School of Physics and Electronics, Hunan University, Changsha, People's Republic of China. [82]Key Laboratory of Nuclear Physics and Ion-beam Application (MOE), Fudan University, Shanghai, People's Republic of China. [83]Institute of Modern Physics, Fudan University, Shanghai, People's Republic of China. [84]Present address: Goethe University Frankfurt, Frankfurt am Main, Germany. [85]Guangdong Provincial Key Laboratory of Nuclear Science, South China Normal University, Guangzhou, People's Republic of China. [86]Institute of Quantum Matter, South China Normal University, Guangzhou, People's Republic of China. [87]Present address: Institute of Physics and Technology, Ulaanbaatar, Mongolia. [88]Present address: Department of Physics, Harvard University, Cambridge, MA, USA. [89]Present address: NRC "Kurchatov Institute", PNPI, Gatchina, Russia. [90]Key Laboratory for Particle Physics, Astrophysics and Cosmology (MOE), Shanghai, People's Republic of China. [91]Shanghai Key Laboratory for Particle Physics and Cosmology, Shanghai, People's Republic of China. [92]Institute of Nuclear and Particle Physics, Shanghai Jiao Tong University, Shanghai, People's Republic of China.

## Methods

### Monte Carlo simulation

For the selection and optimization of the final event sample, estimation of background sources as well as normalization for the fit method, Monte Carlo simulations have been used. The simulation of the BESIII detector is implemented in the simulation software GEANT4[50,51]. GEANT4 takes into account the propagation of the particles in the magnetic field and particle interactions with the detector material. The simulation output is digitized, converting energy loss to pulse heights and points in space to channels. In this way the Monte Carlo digitized data have the same format as the experimental data. The production of the $J/\psi$ is simulated by the Monte Carlo event generator KKMC[52]. Particle decays are simulated using the package BesEvtGen[53,54,55], where the properties of mass, branching ratios and decay lengths come from the world-averaged values[26]. We find that although the mass of the $\Lambda$ in our data agrees with the established value, that of the $\Xi$ is 95 keV/$c^2$ above the central value of the world average[26], $m_{\Xi,\text{PDG}} = 1{,}321.71 \pm 0.07$ MeV/$c^2$ (PDG, Particle Data Group). Hence, we have adjusted the input mass value in the simulation accordingly[55].

The signal channels used for optimization and consistency checks are implemented with the helicity formalism and with parameter values in close proximity to the results presented in Table 1.

### Selection criteria

The data were accumulated during two run periods, in 2009 and 2012, where the later set is approximately five times larger than the earlier. For the analysis all charged final-state particles have to be reconstructed. The main drift chamber of the BESIII experimental set-up is used for reconstructing the charged-particle tracks. At least three positively and three negatively charged tracks are required, each track fulfilling the condition that $|\cos\theta_{\text{LAB}}| < 0.93$, where $\theta_{\text{LAB}}$ is the polar angle with respect to the positron beam direction. The momentum distributions of protons and pions from the signal process are well separated and do not overlap, as shown in Extended Data Fig. 1. Therefore a simple momentum criterion suffices for particle identification: $p_{\text{pr}} > 0.32$ GeV/$c$ and $p_{\pi} < 0.30$ GeV/$c$ for protons and pions, respectively. The probability of misidentifying a proton (antiproton) for a $\pi^+$ ($\pi^-$) is 0.17% (0.18%). Only events with at least one proton, one antiproton, two negatively and two positively charged pions are saved for further analysis. Each $\Xi$ decay chain is reconstructed separately, and is here described for the sequence $\Xi^- \to \Lambda\pi^- \to p\pi^-\pi^-$. To find the correct $\Xi^-$ and $\Lambda$ particles all proton and $\pi^-$ candidates are combined together. The $\Lambda$ and $\Xi^-$ particles are reconstructed through vertex fits by first combining the $p\pi_i^-$ pair to form a $\Lambda$ and then the $\Lambda\pi_j^-$ ($i \neq j$) pair to form a $\Xi^-$. The fits take into account the nonzero flight paths of the hyperons, which can give rise to different production and decay points. All vertex fits must converge and the combination that minimizes $((m_{p\pi\pi} - m_\Xi)^2 + (m_{p\pi} - m_\Lambda)^2)^{1/2}$, where $m_\Xi$ and $m_\Lambda$ are the nominal masses and $m_{p\pi\pi}$ ($m_{p\pi}$) is the mass of the candidate $\Xi$ ($\Lambda$), is retained for further analysis. The same procedure is performed for the $\overline{\Xi}$ decay chain. For each decay chain the probability that the pions from the $\Xi \to \Lambda\pi$ and $\Lambda \to p\pi$ decays are wrongly assigned is found to be 0.51% and 0.49% for $\pi^+$ and $\pi^-$, respectively, which is negligible for the analysis. The $m_{\Lambda\pi^-}$ versus $m_{\overline{\Lambda}\pi^+}$ scatter plot is shown in Extended Data Fig. 2. A four-constraint kinematic fit requiring energy and momentum conservation (4C) is imposed on the $e^+e^- \to J/\psi \to \Xi^-\overline{\Xi}^+$ system, and only events where $\chi^2_{4C} < 100$ are retained for further analysis. The kinematic fit is effective for removing the background processes $e^+e^- \to J/\psi \to \gamma\eta_c \to \gamma\Xi^-\overline{\Xi}^+$ and $e^+e^- \to J/\psi \to \Xi(1530)^-\overline{\Xi} \to \pi^0\Xi^-\overline{\Xi}^+$ (and its charge conjugate), which have the same charged final-state topology as the signal channel, but contain extra neutral particles.

The invariant masses of the $p\pi^-$ and $\overline{p}\pi^+$ pairs are also required to fulfil $|m_{p\pi} - m_{\Lambda,\text{peak}}| < 11.5$ MeV/$c^2$, where $m_{\Lambda,\text{peak}}$ is the peak position of the $\Lambda$ mass distribution. A similar mass window criterion, optimized to remove the broad resonance $\Sigma^-(1385)\overline{\Sigma}^+(1385)$ background contribution, is imposed on the $\Xi$ particle, $|m_{\Lambda\pi} - m_{\Xi,\text{peak}}| < 11.0$ MeV/$c^2$.

The decay length is defined as the distance between the point of origin and the decay position of the decaying $\Lambda$ or $\Xi$ particle. If the hyperon momentum points oppositely to the direction from the collision to the decay point, then the decay length becomes negative in the vertex-fit algorithm. These events are removed from the sample.

Differences between experimental data and Monte Carlo simulations are observed for large polar angles. This discrepancy induces a systematic bias on the parameter values. This bias can, however, be reduced to a negligible level by requiring $|\cos\theta| < 0.84$. The $\Xi$ scattering angle $\theta$ is defined in the main text.

After applying all aforementioned selection criteria, 73,244 $\Xi^-\overline{\Xi}^+$ candidates remain in the final sample. This is shown in Extended Data Fig. 3. The number of remaining background events are estimated to be $199 \pm 17$. The background contribution has a marginal effect on the results at this precision and is therefore neglected.

### Definition of the helicity systems

In the $e^+e^- \to \Xi^-\overline{\Xi}^+$, $\Xi^- \to \Lambda\pi^-$, $\Lambda \to p\pi^-$, $\overline{\Xi}^+ \to \overline{\Lambda}\pi^+$, $\overline{\Lambda} \to \overline{p}\pi^+$ process, the 'master coordinate system', denoted $\mathcal{R}$, is defined in the $e^+e^-$ centre-of-momentum system. In this system, we define the unit vector $\hat{\mathbf{z}}$ in the direction of the positron momentum. The coordinate system $\mathcal{R}_\Xi$ is then defined in the rest frame of the $\Xi^-$ baryon, with the $z$ axis along the unit vector $\hat{\mathbf{z}}_\Xi$ defined by the direction of the $\Xi^-$ momentum in the $\mathcal{R}$ system. A Cartesian coordinate system with $\hat{\mathbf{x}}_\Xi$ and $\hat{\mathbf{y}}_\Xi$ unit vectors is defined as

$$\hat{\mathbf{x}}_\Xi = \frac{\hat{\mathbf{z}} \times \hat{\mathbf{z}}_\Xi}{|\hat{\mathbf{z}} \times \hat{\mathbf{z}}_\Xi|} \times \hat{\mathbf{z}}_\Xi, \quad \hat{\mathbf{y}}_\Xi = \frac{\hat{\mathbf{z}} \times \hat{\mathbf{z}}_\Xi}{|\hat{\mathbf{z}} \times \hat{\mathbf{z}}_\Xi|}. \tag{8}$$

The helicity system $\mathcal{R}_{\overline{\Xi}}$ is defined in the same way in the $\overline{\Xi}^+$ rest frame, and because $\hat{\mathbf{z}}_{\overline{\Xi}} = -\hat{\mathbf{z}}_\Xi$, the axes $\hat{\mathbf{x}}_{\overline{\Xi}} = \hat{\mathbf{x}}_\Xi$ and $\hat{\mathbf{y}}_{\overline{\Xi}} = -\hat{\mathbf{y}}_\Xi$. The system $\mathcal{R}_\Lambda$ is defined in the rest frame of the $\Lambda$, with the $\hat{\mathbf{z}}_\Lambda$ pointing in the direction of the $\Lambda$ momentum in the $\mathcal{R}_\Xi$ system. A new Cartesian coordinate system is then defined by the unit vectors $\hat{\mathbf{x}}_\Lambda$ and $\hat{\mathbf{y}}_\Lambda$

$$\hat{\mathbf{x}}_\Lambda = \frac{\hat{\mathbf{z}}_\Xi \times \hat{\mathbf{z}}_\Lambda}{|\hat{\mathbf{z}}_\Xi \times \hat{\mathbf{z}}_\Lambda|} \times \hat{\mathbf{z}}_\Lambda, \quad \hat{\mathbf{y}}_\Lambda = \frac{\hat{\mathbf{z}}_\Xi \times \hat{\mathbf{z}}_\Lambda}{|\hat{\mathbf{z}}_\Xi \times \hat{\mathbf{z}}_\Lambda|}. \tag{9}$$

In the same way, the system $\mathcal{R}_{\overline{\Lambda}}$ can be derived, and hence there is a unique definition of the orientations of the coordinate systems $\mathcal{R}_\Xi$, $\mathcal{R}_{\overline{\Xi}}$, $\mathcal{R}_\Lambda$ and $\mathcal{R}_{\overline{\Lambda}}$ used in the analysis.

### The maximum log-likelihood fit procedure

The global fit is performed on the data through the joint angular distribution. For $N$ events the likelihood function is given by

$$\mathcal{L}(\boldsymbol{\xi}_1, \boldsymbol{\xi}_2, \ldots, \boldsymbol{\xi}_N; \boldsymbol{\omega}) = \prod_{i=1}^N \mathcal{P}(\boldsymbol{\xi}_i; \boldsymbol{\omega})$$
$$= \prod_{i=1}^N \frac{\mathcal{W}(\boldsymbol{\xi}_i; \boldsymbol{\omega})\varepsilon(\boldsymbol{\xi}_i)}{\mathcal{N}(\boldsymbol{\omega})}, \tag{10}$$

where $\varepsilon(\boldsymbol{\xi})$ is the efficiency, $\mathcal{W}(\boldsymbol{\xi}; \boldsymbol{\omega})$ is the weight as specified in equation (6), and the normalization factor $\mathcal{N}(\boldsymbol{\omega}) = \int \mathcal{W}(\boldsymbol{\xi}; \boldsymbol{\omega})\varepsilon(\boldsymbol{\xi})d\boldsymbol{\xi}$. The normalization factor is approximated as $\mathcal{N}(\boldsymbol{\omega}) \approx \frac{1}{M}\sum_{j=1}^M \mathcal{W}(\boldsymbol{\xi}_j; \boldsymbol{\omega})$, using $M$ Monte Carlo events $\boldsymbol{\xi}_j$ generated uniformly over phase space, propagated through the detector and reconstructed in the same way as data. $M$ is chosen to be much larger than the number of events in data $N$; our results exploit a simulation sample where $M/N \approx 35$. By taking the natural logarithm of the joint probability density, the efficiency function can be separated and removed as it only affects the overall log-likelihood

normalization and is not dependent on the parameters in **ω**. To determine the parameters, the Minuit package from the CERN library is used[56]. The minimized function is given by $S = -\ln(\mathcal{L})$. The operational conditions were slightly different for the 2009 and 2012 datasets, most notably in the nominal value of the magnetic field. For this reason, the likelihoods are constructed separately for the two different run periods.

The results of the simultaneous fit are shown in Table 1. Those results that depend on combinations of decay parameters account for the correlations between the parameters. The correlation coefficients between the decay parameters are given in Extended Data Table 1. Assuming that CP symmetry is conserved we find $\langle\alpha_\Lambda\rangle = 0.760 \pm 0.006 \pm 0.003$ and $\langle\alpha_\Xi\rangle = -0.373 \pm 0.005 \pm 0.002$, where the latter result is in disagreement with the current standard value[26] $\alpha_\Xi = -0.401 \pm 0.010$. The parameter $\alpha_\Xi$ has previously only been measured indirectly via the product $\alpha_\Xi\alpha_\Lambda$ and the assumed value of $\alpha_\Lambda$. The current standard value of $\alpha_\Lambda = 0.732 \pm 0.014$ is an average based on the two incompatible results of BESIII and the re-analysed CLAS data[4,5], and in disagreement with the value found in this analysis. By contrast, our measured value for the product $\langle\alpha_\Xi\rangle\langle\alpha_\Lambda\rangle = -0.284 \pm 0.004 \pm 0.002$ is compatible with the world average[26] $\alpha_\Xi\alpha_\Lambda = -0.294 \pm 0.005$.

### Systematic uncertainties

The systematic uncertainties are assigned by performing studies related to the kinematic fit, the $\Lambda$ and $\Xi$ mass window requirements, the $\Lambda$ and $\Xi$ decay length selection, and a combined test on the $\Xi\bar{\Xi}$ fit reconstruction with the $p$, $\pi$ main drift chamber track reconstruction efficiency. Searches of systematic effects are tested by varying the criteria above and below the main selection. For each test, $i$, the parameter values are re-obtained, $\omega_{\text{sys},i}$ and the changes evaluated compared to the central values, $\Delta_i = |\omega - \omega_{\text{sys},i}|$. Also calculated are the uncorrelated uncertainties $\sigma_{\text{uc},i} = \sqrt{|\sigma_\omega^2 - \sigma_{\omega,\text{sys},i}^2|}$, where $\sigma_\omega$ and $\sigma_{\omega,\text{sys},i}$ correspond to the fit uncertainties of the main and systematic test results, respectively. If the ratio $\Delta_i/\sigma_{\text{uc},i}$ shows a trending behaviour and larger than two this is attributed to a systematic effect[57,58]. For each systematic effect the corresponding uncertainty is evaluated. The assigned systematic uncertainties are given in Extended Data Tables 2–4, where the individual systematic uncertainties are summed in quadrature.

1. **Estimator.** To test if the method produces systematically biased results, a large Monte Carlo data sample is produced with production and decay distributions corresponding to those of the fit results to the data sample (~10 times the experimental data). The simulated data are divided into subsamples with equal number of events as the experimental sample, and run through the fit procedure. The obtained fit parameters and uncertainties are found to be consistent within one standard deviation of the generated parameter values and hence no bias is detected.

2. **Kinematic fit.** The systematic differences from the kinematic fit are tested by varying the kinematic fit $\chi^2$ value from 40 to 200, with an increment of 20 in each step. Significant effects are seen for the parameters $\Delta\Phi$, $\phi$ and $\bar{\phi}$ when $\chi^2_{\text{4C}} > 100$. For $\chi^2_{\text{4C}} < 100$ systematic deviations occur for $\alpha_\Xi$ and $\bar{\alpha}_\Lambda$. The difference in track resolution between data and Monte Carlo is the probable cause for these changes in the parameter values. The systematic uncertainty is assigned to be the average difference of the main result to a lower and upper limit, determined to be at $\chi^2_{\text{4C}} = 60$ and 200, respectively.

3. **$\Lambda$ and $\Xi$ mass window selection.** Possible systematic effects due to the $\bar{\Lambda}\Lambda$, $\bar{\Lambda}$, $\Xi^-$ and $\bar{\Xi}^+$ mass windows are investigated by varying the selection criteria between 2 and 30 MeV/$c^2$ and 2 and 20 MeV/$c^2$ for the $\Lambda/\bar{\Lambda}$ and $\Xi^-/\bar{\Xi}^+$ candidates, respectively. For the $\Lambda$ selection systematic deviations are seen for decreasing mass windows. The uncertainty is assigned to be the difference of the nominal result to the result when 95% of the events are included, at $|m_{p\pi} - m_{\Lambda,\text{peak}}| < 6.9$ MeV/$c^2$. For the $\Xi^-$ and $\bar{\Xi}^+$ mass windows, significant effects are seen for the parameters $\alpha_{J/\psi}$, $\alpha_\Lambda$ and $\phi_\Xi$. The systematic uncertainties for these parameters are assigned to be the difference of the main

result and the results obtained one standard deviation lower than the main selection window, estimated from the $m_{\Lambda\pi}$ line shape uncertainty.

4. **$\Lambda$ and $\Xi$ decay length.** Possible systematic effects related to the $\Lambda$ and $\Xi$ lifetimes are studied by varying the decay length selection criteria for the $\Lambda$ and $\Xi$ candidates. For $\Xi$ no strong trending behaviours are seen, but for $\Lambda$ a dependence is seen for the asymmetry parameters $\alpha_\Lambda$ and $\bar{\alpha}_\Lambda$, which is accounted for in the final systematic uncertainty.

5. **The combined efficiency of $\Xi^-\bar{\Xi}^+$ reconstruction and $p$, $\pi^-$ tracking.** For the study of systematic effects related to the tracking and the $\Lambda$ and $\Xi$ reconstruction it is assumed that the combined efficiency for proton, antiproton and $\pi^\pm$ depends only on the polar angle $\cos\theta_{\text{LAB}}$ and the transverse momentum, $p_{\text{T}}$. To study the tracking efficiency the fitted probability density function is modified by allowing for arbitrary efficiency corrections as a function of $\cos\theta_{\text{LAB}}$ and $p_{\text{T}}$ for each particle type in an iterative procedure. The correction procedure is repeated until the maximum log-likelihood is stable within ln(2) between two successive iterations. The difference between the fit results with and without the tracking correction is assigned as the systematic uncertainty.

6. **The $\cos\theta$ scattering angle.** From comparing data to Monte Carlo simulation a discrepancy is seen for charged tracks with polar angles $|\cos\theta_{\text{LAB}}| > 0.84$. The discrepancy is also seen to have a notable effect on some of the decay parameters. The effect can be isolated by removing only the events where $|\cos\theta| > 0.84$. Although the observed data–simulation differences are removed by requiring that $|\cos\theta| < 0.84$, residual systematic effects are observed for $\langle\alpha_\Xi\rangle$ and $\langle\alpha_\Xi\rangle\langle\alpha_\Lambda\rangle$, which are included in the systematic uncertainty.

7. **Dataset consistency.** When comparing the statistically independent results of the 2009 and 2012 datasets, all parameters are found to agree within two standard deviations. As there is no evidence of systematic bias, no uncertainty is assigned associated with possible dataset differences.

## Data availability

The data points displayed in the plots within this paper are available on request to besiii-publications@ihep.ac.cn.

## Code availability

All algorithms used for data analysis and simulation are archived by the authors and are available on request to besiii-publications@ihep.ac.cn.

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

**Acknowledgements** The BESIII collaboration thanks the staff of BEPCII and the IHEP Computing Centre for their strong support. This work is supported in part by the National Key Research and Development Program of China under contract nos 2020YFA0406300 and 2020YFA0406400; the National Natural Science Foundation of China (NSFC) under contract nos 11625523, 11635010, 11735014, 11822506, 11835012, 11905236, 11935015, 11935016, 11935018, 12075107 and 11961141012; the Chinese Academy of Sciences (CAS) Large-Scale Scientific Facility Program; Joint Large-Scale Scientific Facility Funds of the

NSFC and CAS under contract nos U1732263 and U1832207; the CAS Key Research Program of Frontier Sciences under contract nos QYZDJ-SSW-SLH003 and QYZDJ-SSW-SLH040; the 100 Talents Program of CAS; the CAS President's International Fellowship Initiative (PIFI) programme; INPAC and Shanghai Key Laboratory for Particle Physics and Cosmology; the Shanghai Pujiang Program (20PJ1401700); the ERC under contract no. 758462; the European Union Horizon 2020 research and innovation programme under Marie Skłodowska-Curie grant agreement no. 894790; the German Research Foundation (DFG) under contract nos 443159800, Collaborative Research Center (CRC) 1044, FOR 2359, GRK 214; the Istituto Nazionale di Fisica Nucleare, Italy; the Ministry of Development of Turkey under contract no. DPT2006K-120470; the National Science and Technology fund; the Olle Engkvist Foundation under contract no. 200-0605; STFC (United Kingdom); The Knut and Alice Wallenberg Foundation (Sweden) under contract no. 2016.0157; The Royal Society, UK under contract nos DH140054 and DH160214; the Swedish Research Council; the US Department of Energy under contract nos DE-FG02-05ER41374 and DE-SC-0012069; and the National Science Centre (Poland) under contract no. 2019/35/O/ST2/02907.

**Author contributions** All authors have contributed to the publication, being variously involved in the design and the construction of the detectors, in writing software, calibrating sub-systems, operating the detectors and acquiring data and finally analysing the processed data.

**Funding** Open access funding provided by Uppsala University.

**Competing interests** The authors declare no competing interests.

**Additional information**
**Correspondence and requests for materials** should be addressed to besiii-publications@ihep.ac.cn.

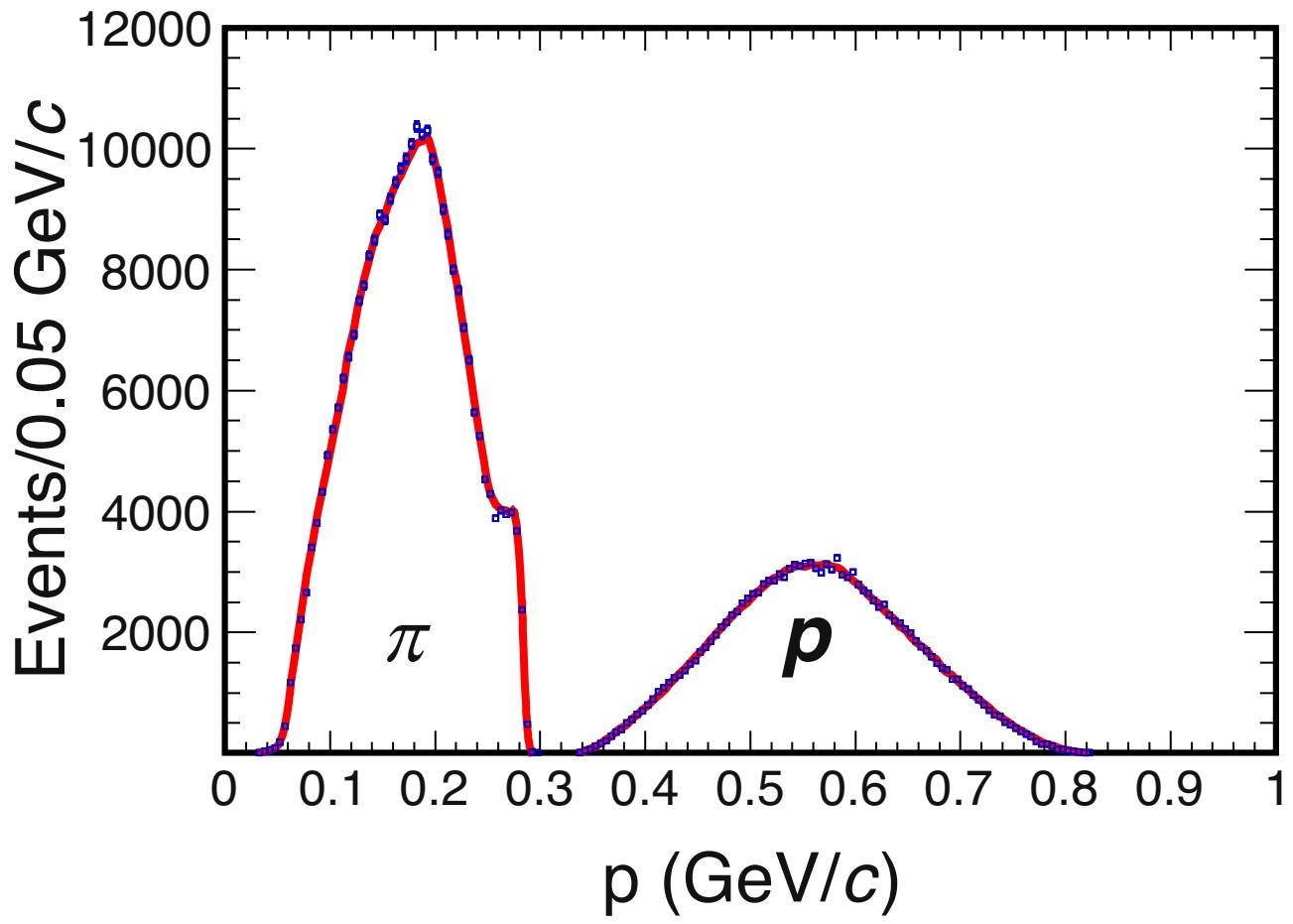

**Extended Data Fig. 1 | Final-state particle momenta of pions and protons for the decay process** $J/\psi \to \Xi^- \overline{\Xi}{}^+ \to \Lambda \pi^- \overline{\Lambda} \pi^+ \to p\pi^- \pi^- \overline{p} \pi^+ \pi^+$**.** The non-overlapping momentum ranges of the protons and pions allow for a straightforward assignment of particle species. The blue boxes and the red solid line denote the experimental and simulated data, respectively.

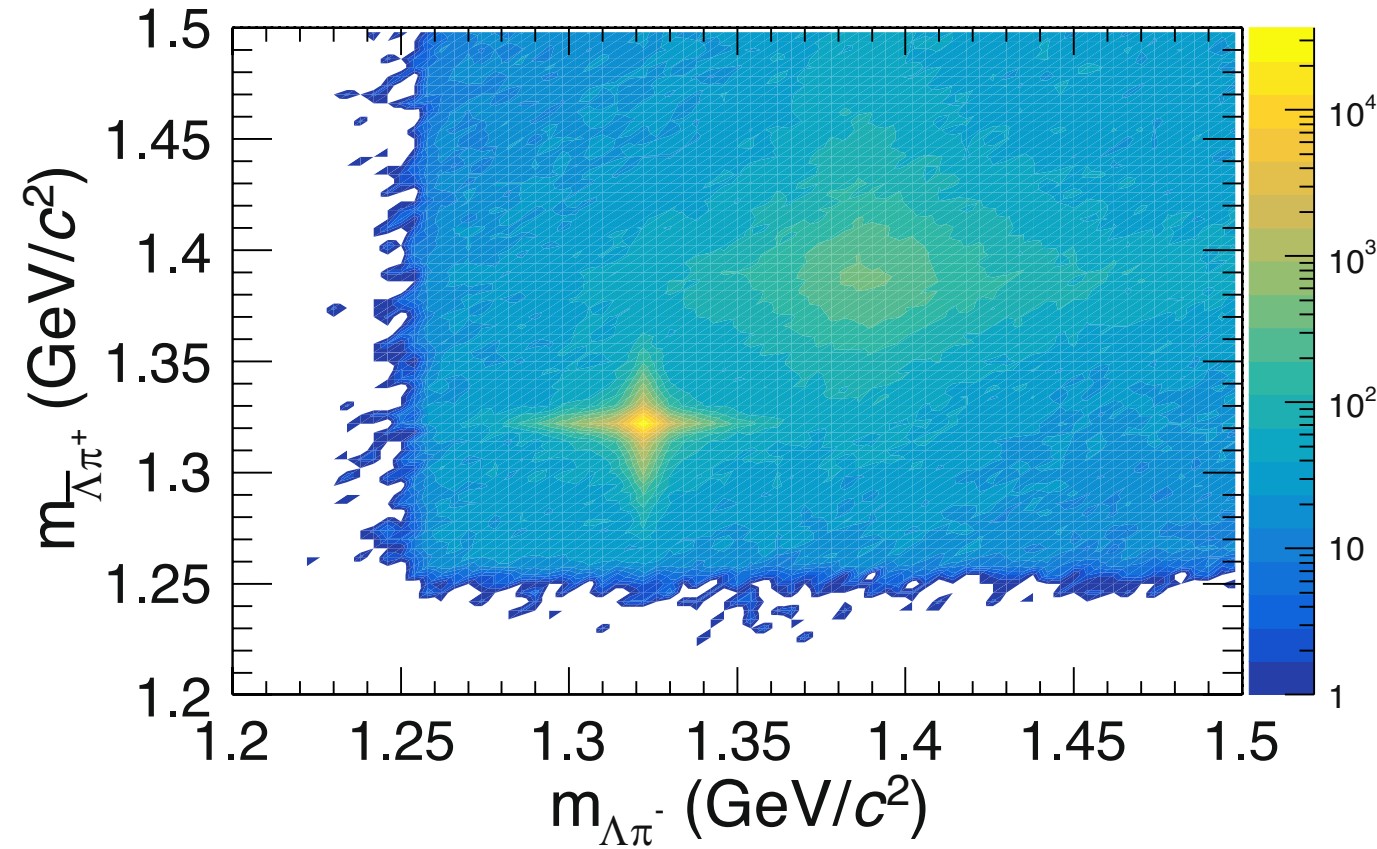

**Extended Data Fig. 2 | Invariant mass distributions of the $\Xi^-$ and $\bar{\Xi}^+$ signal candidates.** Distribution of the invariant masses $m_{\Lambda\pi^-}$ versus $m_{\bar{\Lambda}\pi^+}$. The $\Xi^-\bar{\Xi}^+$ candidates appear as an enhancement around $m_{\Lambda\pi^-} = m_{\bar{\Lambda}\pi^+} = 1.32$ GeV/$c^2$. The structure at $m_{\Lambda\pi^-} = m_{\bar{\Lambda}\pi^+} = 1.39$ GeV/$c^2$ is from the reaction $J/\psi \to \Sigma(1385)^-\bar{\Sigma}(1385)^+$.

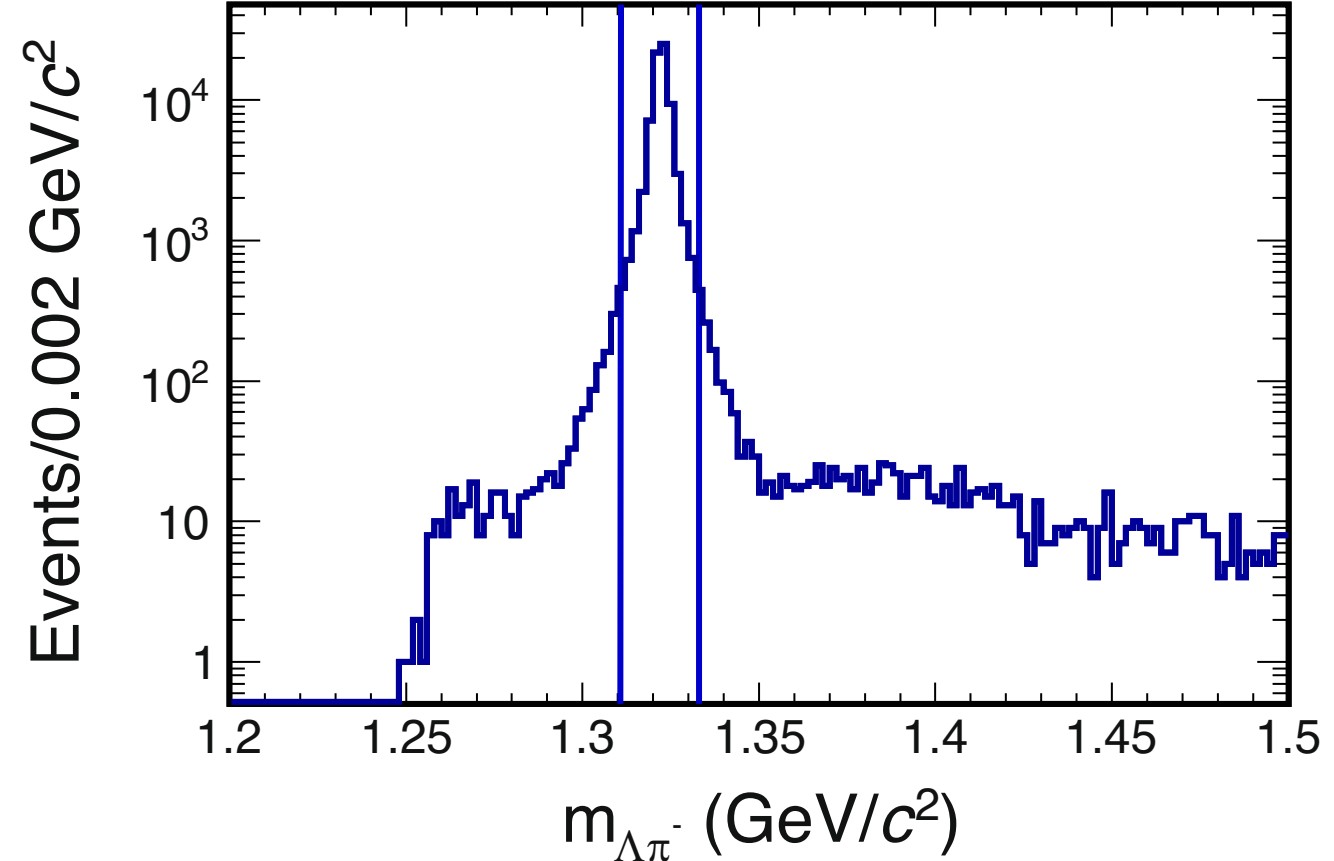

**Extended Data Fig. 3 | Invariant mass distribution of the $\Xi^-$ signal candidates before the final selection criterion.** The $m_{\Lambda\pi^-}$ distribution, in log scale, for the BESIII data sample before the $\Lambda\pi^-$ mass window has been applied. The final requirement selects the events between the two lines. The total number of events in the distribution is 76,523.

**Extended Data Table 1 | Correlation coefficients for the production and asymmetry decay parameters**

|  | $\alpha_\psi$ | $\Delta\Phi$ | $\alpha_\Xi$ | $\phi$ | $\alpha_\Lambda$ | $\overline{\alpha}_\Xi$ | $\overline{\alpha}_\Lambda$ | $\overline{\phi}_\Xi$ |
|---|---|---|---|---|---|---|---|---|
| $\alpha_\psi$ | 1.0 | 0.414 | -0.008 | -0.006 | -0.107 | 0.014 | 0.120 | 0.003 |
| $\Delta\Phi$ |  | 1.0 | -0.016 | 0.016 | -0.133 | 0.008 | 0.138 | -0.029 |
| $\alpha_\Xi$ |  |  | 1.0 | -0.000 | 0.280 | 0.024 | 0.071 | 0.010 |
| $\phi_\Xi$ |  |  |  | 1.0 | -0.002 | -0.010 | -0.010 | 0.013 |
| $\alpha_\Lambda$ |  |  |  |  | 1.0 | 0.070 | 0.401 | 0.014 |
| $\overline{\alpha}_\Xi$ |  |  |  |  |  | 1.0 | 0.269 | 0.001 |
| $\overline{\alpha}_\Lambda$ |  |  |  |  |  |  | 1.0 | 0.006 |
| $\overline{\phi}_\Xi$ |  |  |  |  |  |  |  | 1.0 |

**Extended Data Table 2 | Contributing systematic uncertainties, and the sum in quadrature**

| $\times 10^2$ | $\alpha_\psi$ | $\Delta\Phi$ | $\alpha_\Xi$ | $\overline{\alpha}_\Xi$ | $\alpha_\Lambda$ | $\overline{\alpha}_\Lambda$ | $\phi_\Xi$ | $\overline{\phi}_\Xi$ |
|---|---|---|---|---|---|---|---|---|
| Statistical | 1.2 | 4.6 | 0.70 | 0.70 | 1.05 | 1.06 | 1.91 | 1.93 |
| Kin. fit | 0.36 | 1.5 | 0.18 | 0.17 | 0.21 | 0.43 | 0.77 | 0.44 |
| mass win $\Lambda$ | 0.44 | 0.44 | 0.07 | 0.02 | 0.56 | 0.33 | 0.17 | 0.46 |
| mass win $\Xi$ | 0.25 | - | - | - | 0.36 | - | 0.46 | - |
| dec. length $\Lambda$ | - | - | - | - | 0.30 | 0.40 | - | - |
| Track. eff. | 0.80 | 0.41 | 0.27 | 0.05 | 0.21 | 0.14 | 0.16 | 0.16 |
| Sum syst. | 1.0 | 1.6 | 0.33 | 0.18 | 0.79 | 0.69 | 0.93 | 0.66 |

First row: statistical uncertainty as reference. The uncertainties of $\Delta\Phi$ and $\phi$ are given in radians. All values multiplied by a factor $10^2$, as indicated at top left.

**Extended Data Table 3 | Contributing systematic uncertainties to CP tests, and the sum in quadrature**

| $\times 10^2$ | $A_{\Lambda,\mathrm{CP}}$ | $A_{\Xi,\mathrm{CP}}$ | $\Delta\phi_{\mathrm{CP}}$ (rad) | $\delta_P - \delta_S$ (rad) | $\zeta_P - \zeta_S$ (rad) |
|---|---|---|---|---|---|
| Statistical | 1.17 | 1.34 | 1.37 | 3.3 | 3.4 |
| Kin. fit | 0.32 | 0.47 | 0.16 | 1.3 | 0.4 |
| mass win. $\Lambda$ | 0.59 | 0.07 | 0.14 | 0.8 | 0.4 |
| mass win. $\Xi$ | 0.38 | - | 0.20 | 0.7 | 0.5 |
| dec. length $\Lambda$ | 0.46 | - | - | - | - |
| Track. eff. | 0.05 | 0.29 | 0.003 | 0.4 | $2 \cdot 10^{-3}$ |
| Sum syst. | 0.90 | 0.56 | 0.29 | 1.7 | 0.75 |

First row: statistical uncertainty as reference. All values multiplied by a factor $10^2$, as indicated at top left.

**Extended Data Table 4 | Contributing systematic uncertainties to average values of decay parameters, and the sum in quadrature**

| $\times 10^2$ | $\langle \alpha_\Xi \rangle$ | $\langle \alpha_\Lambda \rangle$ | $\langle \phi \rangle$ (rad) | $\langle \alpha_\Xi \rangle \cdot \langle \alpha_\Lambda \rangle$ |
|---|---|---|---|---|
| Statistical | 0.49 | 0.58 | 1.35 | 0.38 |
| Kin. fit | 0.09 | 0.19 | 0.54 | 0.02 |
| mass win. $\Lambda$ | 0.05 | 0.12 | 0.31 | $< 10^{-2}$ |
| mass win. $\Xi$ | - | 0.07 | 0.26 | 0.04 |
| $\cos \theta_{\Xi,\text{c.m.}}$ | 0.12 | - | - | 0.13 |
| Track. eff. | 0.16 | 0.17 | 0.16 | 0.06 |
| Sum syst. | 0.22 | 0.29 | 0.69 | 0.21 |

First row: statistical uncertainty as reference. All values multiplied by a factor of $10^2$, as indicated at top left.