## [Peer Review File · Nature]

Manuscript Title: Probing CP symmetry and weak phases with entangled double-strange baryons

Reviewer Comments & Author Rebuttals

Reviewer Reports on the Initial Version:

Referees' comments:

Referee #1 (Remarks to the Author):

The manuscript concerns tests of CP violation from the sequential decays of entangled doubly strange baryons emerging from J/ψ decay. The key results of this paper are the reported measurement of three different CP-violating asymmetries, associated with charged Ξ baryons, their subsequent decays, as well as with the Λ baryon and its antiparticle. The first two quantities are claimed to be measured for the first time; more precisely (as the authors note in discussion of earlier HyperCP results) the first result is measured uniquely for the first time.

The paper employs a new and powerful method developed by Adlarson and Kupsc (their ref.2) for the first time.

The authors show that the subsequent asymmetry from the entangled baryon method they employ can be manipulated to yield a quantity $\langle\phi\rangle$, finding it to be of comparable precision to that from the polarized baryon method used by the HyperCP collaboration, albeit with a data set a thousand times smaller.

The methodologies, use of statistics, and treatment of uncertainties appear to be appropriate, and the conclusions are also appropriate. I also find the writing to be lucid and clear.

I do find fault with the references to earlier/other work; this is important to putting the current work in the proper context, though the method employed is certainly still original.

I detail my minor criticisms here:

line 29, since the CKM matrix elements $V_{\{tb,cs,ud\}} \sim O(1)$, it is usually said that the SM mechanism of CP violation is "too special" (rather than "too small") to give sizeable effects, because all three generations of quarks must participate to give a nonzero effect.

line 31, regarding "no CP-violating effects beyond the SM have been observed" --- I daresay this statement refers to measured deviations in excess of 5σ in significance. There are long-standing hints of departures from the Standard Model. Note, e.g., the long-standing B to πK puzzle (Buras et al., 2003), which has recently been sharpened by a measurement by LHCb, <https://arxiv.org/abs/2012.12789> .

line 38, I am not sure ϵ_s'/ϵ_s provides the most stringent BSM constraint for strange hadrons, because ϵ_s'/ϵ_s is so challenging to compute theoretically. Note, Fig. 12.2 of <https://pdg.lbl.gov/2019/reviews/rpp2019-rev-ckm-matrix.pdf> which shows the constraint that comes from the study of B_s mixing (note Δm_s). The qualifier "light" before "strange" would make what they say indisputable. This would also be helpful in regards to line 41, because there are several examples of CP-violating observables in B-meson decays with strange hadrons that do not rely on strong interaction effects.

line 43,43, nuclear and neutron beta decay studies do not separate strong from weak effects through the use of spin. Rather, the use of spin allows the identification of decay correlations whose pattern in size depend on SM inputs in a falsifiable way.

I was asked by the editor to note, if the paper should be acceptable for publication, what its most outstanding features might be. I find this paper to be important, particularly in regards to the insights into the nature of CP violation the study of J/ψ decays and its daughter particles with the entangled baryon method that future studies should bring. However, unlike this LHCb result:

<https://www.nature.com/articles/nature14474>

the current paper gives no striking new insights into the mechanism of CP violation. On the other hand, I find the outcomes of this paper more important than Ref. 4, which was published in Nature Physics. To my mind the most striking finding in the current paper is the demonstrated improvement that the use of entanglement, the imprimatur of quantum mechanics, gives in leveraging the sensitivity of tests of CP violation. This last point seems to me to be of high interest, possibly earning this paper acceptance in Nature, even if I do not find the case a compelling one.

Referee #2 (Remarks to the Author):

The paper reports the measurement of CP parameters using hyperons, some of the reported parameters have been measured for the first time. The method employed is novel and relies on the spin entanglement between the doubly strange baryons. It is appealing to see that the method gives a higher sensitivity to the observables of interest when comparing to other approaches. The author use data collected at BES III, though they argue in the conclusions that this approach could be employed in other experimental environments.

The paper is well written and even if no new sign of CP violation are found in the analysis, the methodology results remain interesting.

I recommend the publication of this work Nature, once the comments and questions below have been addressed.

General comments and questions :

- From the text it is not clear how the backgrounds are subtracted (page 19). In particular there is no reference to the 187 ± 16 in the Methods section.
- Could there be detection efficiencies differences between the π^+/π^- p^+/p^- that could manifest in

the analysis ?

- Has the angular resolution been investigated ?
- Could you consider adding the equations for definition of the angles in the supplementary material. It will be helpful for other experiments if they were to use the method proposed in the paper.
- Are there BSM models which predict CP violation in hyperons ?
- The quality of the figures, could be improved.
- The format of the references should be reconsidered and keep the subscript for footnotes.

Line 30 : Add references at the end of the statement.

Line 32 : Add references at the end of the statement.

Line 34: This is not valid in B and D decays. If you mean to be specific in the strange sector, please add references.

Line 39 : Could you express the formalism on how the phases contribute to make the point about disentangling the weak/BSM and strong one clear.

Line 46 : Could you clarify why we expect to have a P-wave and an S-wave amplitude?

Line 68/Eq 4 : In which observable is this the leading order of ? (Same comments for Equation 5).

Line 71: Could you elaborate how is this observable complementary to ϵ' (besides that fact that it contains only an s-wave) ?

Line 101 : Add references.

Line 105 : the notation is confusing and gives the impression that cc applied only to the Lambda decay.

Line 144 : How are the remaining background events treated in the likelihood fit?

Line 249 : How is the simulation adjusted?

Line 251 : Can you give an estimate of how large are the simulation samples.

Line 260 : Though the cuts seem to select very pure samples is there any mis-identification that remain ?

Line 269 : Are there multiple candidates? If yes how are they handled?

Line 273 : Does the kinetic fit require that the Lambda mass is set to the PDG value ?

Line 276 : What fraction of the background mentioned here remains in the analysis ?

Line 290 : Does this requirement cost a lot of signal efficiency ?

Line 291 : Could you add a plot to illustrate the final set of data used for the analysis ? How is the number of signal evaluated ?

Do you make the assumption that there is no background remain at this stage ?

Line 295 : Can you provide more information about the efficiency? How it's computed? How do you insure that it

does not distort the results? The treatment of the efficiency does not seem to enter in the lists of systematic uncertainties?

Line 315 : Do you have an explanation why the product agrees with the previous result ?

Line 335 : are the uncertainties also well behaved from this procedure?

Line 337 : Does the variation of the kinetic fit impact the background contamination ? If yes how this is taken into account in each

of the 20 steps ? Same question for the Lambda and Xi mass windows systematic uncertainties.

Figure 1 : are these distribution efficiency corrected?

Figure 1/2 of supplementary material: are both run periods included ?

Referee #3 (Remarks to the Author):

Dear authors,

The paper presents the first application, using the BESSIII data, of a very sensitive test of CP symmetry, in decays of doubly strange hyperions (“Xi”), to test the Standard Model and search for new physics, in the same spirit as the measurements of ϵ'/ϵ in kaons. The method was presented before in references [2] and [26] of the paper draft in 2019, where a sensitivity study had been done, but this paper shows the first application of this method in an analysis of experimental data. The method is based on a complex angular analysis exploiting the spin correlations of the pairs of Xi – anti Xi produced in the decays of J/Psi made at the BESS experiment, in their weak decay chains (Xi \rightarrow Lambda pi ; Lambda \rightarrow proton pi). This angular analysis allows one to disentangle the P-wave from the S-wave contributions, and measure the complete set of decay parameters.

The novelty in the results is the use of this new method on real data, which allows not only very precise CP symmetry tests (improvement of the sensitivity of an order of magnitude), but also to disentangle the weak – (or new physics) parts from the strong interaction parts, and get more precise and direct measurements. This is complementary to the ϵ'/ϵ measurement in kaons, which has uncertainties from strong interactions. The accuracy of the results is dominated by the statistical uncertainty, and the accuracy gets close to the one of the theoretical prediction from the standard model. So even if CP violation in Xi decays is not observed, the measurement is a very precise test of the standard model and paves the way for future experiments like PANDA at FAIR which could reach even better sensitivity, and see new physics if it exists. New parameters on Xi decays have also been measured for the first time, and can be useful for other studies like spectroscopy. Parameters on Lambda decays are also measured very precisely and with a more direct approach, and can be compared to results of other experiments.

Those results deserve being published here: this first application on data of a very promising method, is much more accurate, more direct, and more complete with different “views” (like a tool box) of CP tests compared to the classical CP asymmetry approach. It also provides measurements of interesting new parameters, and a fundamental test of the Standard Model. There are also interesting and precise additional results on the Lambda hyperon that can be compared to other experiments, and the Xi has been studied more deeply.

The methodology is well described (selection of data, observables, likelihood fit, extraction of the parameters), looks robust and it has been validated with full Monte-Carlo simulations (an order of magnitude bigger in statistics compared to the data). The data have clear signals and relatively low backgrounds. Systematics effects are studied in details. The analysis seems robust. There are a couple questions to clarify (see below).

The paper is well written, the references are appropriate, the abstract and summary, introduction and conclusion are clear, even if the analysis is complex. A few improvements are suggested below.

Suggested improvements and questions:

Analysis cross checks and tests:

1) Data fit Quality test for the likelihood fit procedure : did you compare the Chi2 value at the maximum likelihood you obtain with the data, to the distribution of the Chi2 values obtained for the 10 subsamples of the Monte-Carlo (same statistics as data, and parameters close as the ones found in the data), in order to check the fit “quality” in the data (cf lines 330 to 335)?

2) Systematics (cf lines 321 to 327) – Does it mean if you think according to that criteria described there that the shift is due to a statistical fluctuation, you don’t account for that as a systematics? If you had, all the effects added up would have been big compared to the “real” systematics effects? Is that method to evaluate if we have a real systematics described in reference [47,48] fully reliable? Is the cut at 2 sigma optimized?

3) Xi decay length distribution

The decay length can vary, so it is accounted for in the likelihood fit, but not considered as an “interest” parameter. What is the typical average decay length (the lifetime is close to 100 ps)? Have you checked the decay length distribution to be consistent with the Xi decay, without carefully re-measuring the Xi lifetime, but as a test?

Possible improvements in presentation:

4) It is said in the conclusion that this measurement is “complementary” to epsilon’/epsilon measurement in kaons. What do you mean exactly? It is the same spirit, looking for CP violation in strange hadrons decays, to test the standard model and look for beyond standard model physics. The method is different, as the type parameters studied, so the “view” on the CP tests (also clear of strong phase effects), and the strange hadron too... It is complementary in the method and approach or could it shed a different light on possible new physics? Have you examples of possible new physics scenarios that would affect differently the measurements of epsilon’/epsilon and this analysis?

5) Line 155 “allows for three independent CP symmetry tests”. It seems those tests are detailed from lines 156 to line 178. But they are not clearly listed as first test, second test, third test. You can guess that (first test, classical : A_{CP} , second test $\Delta_{\phi_{CP}}^{\Xi}$, third $\text{Chsi}_P - \text{Chi}_S$ weak phases differences, with the last two tests very new) ? But it is not fully highlighted. Also in table (supplementary) 3, systematics uncertainties on the CP tests, in the caption it would be good to remind the reader of the 3 different CP tests with the corresponding parameters (1: A_{CP_Lambda} & A_{CP_Xi} ; 2: $\Delta_{\phi_{CP}}$; 3: weak phase difference but you need to fit also the strong phases together with it)

6) For the results for which there was a previous measurement by another experiment (lots of results are also first measurements), and for which there is a “tension”, but the approach is different, are there plans for more studies or hints of why it could be so?

7) More minor comments

a. Line 69: why is the strong phase difference between P and S the ratio beta over alpha?

b. Line 251: Monte-Carlo parameters close to the parameters measured in data (table 1). It means that initially those values were unknown, and once they were known, the simulation is reweighted or redone, so is it kind of an iterative process?

c. Line 290: is the cut on the Xi polar angle optimized for 0.84? If yes, how?

TYPOS?

Line 334 – “mass windows are investigate” -> investigated?

Line 351 “lower than to the main” -> lower than the main?

Sincerely,

Sandrine Emery-Schrenk

Author Rebuttals to Initial Comments:

Comments to Referees:

Weak phases and CP-symmetry tests in sequential decays of entangled double-strange baryons

Nature manuscript 2021-05-08135

BESIII collaboration

I. Note to referees

Line numbers given by referees in comments and questions refer to the old version of the manuscript. All line numbers in our response refers to the updated manuscript.

II. Referee 1

The manuscript concerns tests of CP violation from the sequential decays of entangled doubly strange baryons emerging from J/ψ decay. The key results of this paper are the reported measurement of three different CP-violating asymmetries, associated with charged Σ baryons, their subsequent decays, as well as with the Λ baryon and its antiparticle. The first two quantities are claimed to be measured for the first time; more precisely (as the authors note in discussion of earlier HyperCP results) the first result is measured uniquely for the first time.

The paper employs a new and powerful method developed by Adlarson and Kupsc (their ref.2) for the first time. The authors show that the subsequent asymmetry from the entangled baryon method they employ can be manipulated to yield a quantity $\langle \phi \rangle$, finding it to be of comparable precision to that from the polarized baryon method used by the HyperCP collaboration, albeit with a data set a thousand times smaller.

The methodologies, use of statistics, and treatment of uncertainties appear to be appropriate, and the conclusions are also appropriate. I also find the writing to be lucid and clear.

I do find fault with the references to earlier/other work; this is important to putting the current work in the proper context, though the method employed is certainly still original. I detail my minor criticisms here:

1. line 29, since the CKM matrix elements $V_{tb,cs,ud} \mathcal{O}(1)$, it is usually said that the SM mechanism of CP violation is "too special" (rather than "too small") to give sizeable effects, because all three generations of quarks must participate to give a nonzero effect.
Response: We changed the sentence. The modified sentence reads: "However, the SM mechanisms are too specific to yield effects of a size that can explain the observed matter-antimatter asymmetry of the Universe [8, 9]. Therefore, ..."
2. line 31, regarding "no CP-violating effects beyond the SM have been observed" – I daresay this statement refers to measured deviations in excess of 5σ in significance. There are long-standing hints of departures from the Standard Model. Note, e.g., the long-standing $B \rightarrow \pi K$ puzzle (Buras et al., 2003), which has recently been sharpened by a measurement by LHCb, <https://arxiv.org/abs/2012.12789>.

Response: We agree that there exist tensions which could point to BSM physics. We softened the statement by omitting the word "meson". The modified sentence reads: "So far, no CP violating effects beyond the SM have been observed in the baryon sector"

3. line 38, I am not sure ϵ'/ϵ provides the most stringent BSM constraint for strange hadrons, because ϵ'/ϵ is so challenging to compute theoretically.

Note, Fig. 12.2 of <https://pdg.lbl.gov/2019/reviews/rpp2019-rev-ckm-matrix.pdf> which shows the constraint that comes from the study of B_s mixing (note Δm_s). The qualifier "light" before "strange" would make what they say indisputable. This would also be helpful in regards to line 41, because there are several examples of CP-violating observables in B-meson decays with strange hadrons that do not rely on strong interaction effects.

Response: We added "light" in front of "strange" as suggested.

4. line 43, nuclear and neutron beta decay studies do not separate strong from weak effects through the use of spin. Rather, the use of spin allows the identification of decay correlations whose pattern in size depend on SM inputs in a falsifiable way.

Response: We changed the paragraph to:

Baryons provide additional information through spin measurements. Known examples involving three-body decays are spin correlations and polarisation in nuclear and neutron β decays [16]. Sequential two-body decays of entangled multi-strange baryon-antibaryon pairs provide another, hitherto unexplored diagnostic tool to separate the strong and the weak phases.

I was asked by the editor to note, if the paper should be acceptable for publication, what its most outstanding features might be. I find this paper to be important, particularly in regards to the insights into the nature of CP violation the study of J/ψ decays and its daughter particles with the entangled baryon method that future studies should bring. However, unlike this LHCb result: <https://www.nature.com/articles/nature14474> the current paper gives no striking new insights into the mechanism of CP violation. On the other hand, I find the outcomes of this paper more important than Ref. 4, which was published in Nature Physics. To my mind the most striking finding in the current paper is the demonstrated improvement that the use of entanglement, the imprimatur of quantum mechanics, gives in leveraging the sensitivity of tests of CP violation. This last point seems to me to be of high interest, possibly earning this paper acceptance in Nature, even if I do not find the case a compelling one.

III. Referee 2

The paper reports the measurement of CP parameters using hyperons, some of the reported parameters have been measured for the first time. The method employed is novel and relies on the spin entanglement between the doubly strange baryons. It is appealing to see that the method gives a higher sensitivity to the observables of interest when comparing to other approaches. The author use data collected at BESIII, though they argue in the conclusions that this approach could be employed in other experimental environments. The paper is well written and even if no new sign of CP violation are found in the analysis, the methodology results remain interesting. I recommend the publication of this work Nature, once the comments and questions below have been addressed.

General comments and questions :

1. a) From the text it is not clear how the backgrounds are subtracted (page 19). b) In particular there is no reference to the 187 ± 16 in the Methods section.

Response: a) The background is not subtracted as its contribution to the results are minor. More details are provided in the follow-up comments 17, 26 and 30. In addition, we have provided information about the background in the supplementary material (Methods).

b) When we wrote "More details are given in section Methods" we meant that more details of the analysis procedure are given in Methods, not just the background. We have now clarified this aspect: "More details of the analysis are given in the supplementary Methods section".

2. Could there be detection efficiencies differences between the π^+/π^- p^+/p^- that could manifest in the analysis?

Response: Differences between particle and antiparticles were considered and described in "The combined efficiency of $E^- \bar{E}^+$ reconstruction and $p \pi^-$ tracking", L371-379. In this cross-check, we used separate nuisance parameters for particles and anti-particles. This test would reveal possible data-MC differences that were not accounted for. More details are given in item 27. The differences are small and their effect is included in the systematic uncertainty as "Track eff" in Tables 2- 4 in the supplementary material.

3. Has the angular resolution been investigated?

Response: We assume that a diagonal response matrix is a reasonable approximation. We have investigated the validity of this assumption by performing a bias test, described in L343-L349. The test compares results of the fits using MC true and reconstructed variables and shows that the assumption does not introduce bias for the values and uncertainties of the extracted parameters.

4. Could you consider adding the equations for definition of the angles in the supplementary material. It will be helpful for other experiments if they were to use the method proposed in the paper.

Response: In the Methods part of the supplementary material, we have included a paragraph with formulas which define the orientation of the helicity frames. See lines 303-305.

5. Are there BSM models which predict CP violation in hyperons?

Response: We think that general approaches rather than individual models are of larger importance for relating weak, CP-odd phases in hyperon decays to the ϵ' and ϵ values. Such a general approach is presented by J. Tandean in ref. [5]. For a general BSM class, where the CPV effects are dominated by chromomagnetic-penguin operators, a relation is derived between the hyperon and kaon observables. This, as well as other predictions made in the beginning of this century, were motivated by the ongoing HyperCP experiment. We hope the method presented here in conjunction with prospects of future measurements e.g., by PANDA and the planned Super-charm τ experiments, will motivate the theory community to update and extend the predictions. We modified the text in L200-203 as follows:

The contributions to ϵ and ϵ' from hyperon decays on the one hand and kaon decays on the other, are described by different combinations of quark operators. In addition, hyperons provide information on the spin structure of the operators that is not possible to obtain from kaon decays.

6. The quality of the figures, could be improved.

Response: We have updated Figure 2 in the main part of the manuscript.

7. The format of the references should be reconsidered and keep the subscript for footnotes.
Response: We have used the reference format which is suggested by the Nature editors.

8. Line 30: Add references at the end of the statement.
Response: We added reference "Bigi, I. I. & Sanda A. I. CP violation, (2009)."

9. Line 32: Add references at the end of the statement.
Response: We added reference "PDG Review "CP violation in the quark sector" (2019)"

10. Line 34: This is not valid in B and D decays. If you mean to be specific in the strange sector, please add references.

Response: Our intention was to make a general statement concerning direct CP-violation mechanism in line with the discussion in the book on CP-violation by Bigi and Sanda (Ref. 10 in current manuscript). Revealing a CP-violating signal requires at least two amplitudes that involve both CP-even and CP-odd transitions. According to Watson's theorem, decay processes with two or more hadrons in the final state must involve strong final state interactions. The latter constitute the prime candidate to generate CP-even transitions. This mechanism is used in both B and D meson decays (see the recent review Bediaga & Göbel Prog. Part. Nucl. Phys. 114 (2020) 103808 and for example the recent LHCb analysis for B mesons Phys. Rev. Lett. 124 (2020) 031801 and Phys. Rev. D 101, 012006).

We added the Bigi & Sanda (Ref. 10) and Bediaga & Göbel (Ref. 11) references in the manuscript.

11. Line 39: Could you express the formalism on how the phases contribute to make the point about disentangling the weak/BSM and strong one clear.

Response: The observable CP-violation effect is due to interference between at least two amplitudes with different values of CP-odd and CP-even phases. For kaon decays, the two transitions correspond to the isospin $I = 0$ and $I = 2$ states of the final-state pions ($|\Delta I| = 1/2$ and $|\Delta I| = 3/2$ transitions, respectively). Using the notation $A_{2\Delta I, 2I}$, the amplitudes, Eq. (7.18) in Bigi Ref. 10 in updated manuscript, are expressed as

$$A(K^0 \rightarrow \pi^+ \pi^-) = \sqrt{\frac{1}{3}} A_{3,4} \exp(i\xi_{3,4} + i\delta_4) + \sqrt{\frac{2}{3}} A_{1,0} \exp(i\xi_{1,0} + i\delta_0)$$

$$A(K^0 \rightarrow \pi^0 \pi^0) = \sqrt{\frac{2}{3}} A_{3,4} \exp(i\xi_{3,4} + i\delta_4) - \sqrt{\frac{2}{3}} A_{1,0} \exp(i\xi_{1,0} + i\delta_0),$$

where $\xi_{1,0}$ and $\xi_{3,4}$ are weak phases corresponding to the $|\Delta I| = 1/2$ and $|\Delta I| = 3/2$ transitions, respectively. The strong phase shifts in the final-state pion systems with isospin $I = 0$ and $I = 2$ are δ_0 and δ_4 , respectively. For $\epsilon' \neq 0$, both amplitudes $A_{1,0}$ and $A_{3,4}$ must be present. In the above notation, the expression for ϵ' , Eq. (7.34) from Ref. (see Bigi, Ref.10), reads

$$\epsilon' \approx -\frac{i}{\sqrt{2}} \exp(i\delta_4 - i\delta_0) \frac{A_{3,4}}{A_{1,0}} (\xi_{1,0} - \xi_{3,4}). \quad (1)$$

The corresponding expressions for E^- decay are given in the answer to Comment 7 of Reviewer 3. For hyperons, the dominating CP-violation effect is the $|\Delta I| = 1/2$ transition.

12. Line 46: Could you clarify why we expect to have a P-wave and an S-wave amplitude?
Response: When a spin 1/2 hyperon decays into a spin 1/2 baryon and pseudoscalar meson ($Y \rightarrow BP$), conservation of total spin implies that the two final state particles (the

spin 1/2 baryon and the pseudoscalar) can have a relative angular momentum L of 0 or 1. The parity of a weak decay is not conserved so the parity of the final state can be either positive or negative. Knowing that the parity of the final state is given by $(-1)^{L+1}$, it is clear that parity conservation corresponds to a P state and parity violation to an S state.

13. Line 68/Eq 4: In which observable is this the leading order of ? (Same comments for Equation 5).

Response: **General answer:** It is the leading order for the CP-violation effect. The Ξ decays have four amplitudes. Each of the amplitudes can have a weak CP-odd phase. To observe CP-violation, only two amplitudes (two weak phases) are needed. The $\Delta I = 3/2$ amplitudes and the CP-violation effects connected to them are suppressed by a factor of 20 (the exact value of the suppression factor can be determined by comparing the life times and decay properties of Ξ^- and Ξ^0). Contrary to the kaon decays, where $\Delta I = 3/2$ amplitudes are necessary to observe CP-violation, these amplitudes can be neglected in hyperon decays unless a precision better than 5% is required for the value of the CP-odd observables. All theory predictions use this approximation.

Detailed answer: Here we sketch (following ref. [4]) the derivation of Eq. 4 and Eq. 5 including $\Delta I = 3/2$ amplitudes. The amplitude for the $\Xi \rightarrow \Lambda\pi$ decay is

$$A(\Xi^- \rightarrow \Lambda\pi^-) = S + P\sigma \cdot \hat{\mathbf{n}}, \quad (2)$$

where $\hat{\mathbf{n}} = \mathbf{q}/|\mathbf{q}|$ is the direction of the Λ momentum \mathbf{q} in the Ξ rest frame. The S and P amplitudes can be written in the isospin basis as

$$S = S_{1/2}\exp(i\xi_{1/2}^S + i\delta_{I=1}^S) + \frac{1}{2}S_{3/2}\exp(i\xi_{3/2}^S + i\delta_{I=1}^S) \quad (3)$$

$$P = P_{1/2}\exp(i\xi_{1/2}^P + i\delta_{I=1}^P) + \frac{1}{2}P_{3/2}\exp(i\xi_{3/2}^P + i\delta_{I=1}^P), \quad (4)$$

where $S_{\Delta I}(P_{\Delta I})$ and $\xi_{\Delta I}^S(\xi_{\Delta I}^P)$ are the magnitude and the CP-odd phase for the two possible weak transitions changing isospin I by $\Delta I = \frac{1}{2}$ and $\frac{3}{2}$ respectively. The phase shifts $\delta_{I=1}^S$ and $\delta_{I=1}^P$ represent the strong π - Λ scattering in the final state. They are labelled by the isospin $I = 1$ value in the final state. The magnitudes $S_{\Delta I}$ and $P_{\Delta I}$ are real numbers. The amplitudes for $\bar{\Xi}^- \rightarrow \bar{\Lambda}\pi^+$ are

$$\bar{S} = -S_{1/2}\exp(-i\xi_{1/2}^S + i\delta_{I=1}^S) - \frac{1}{2}S_{3/2}\exp(-i\xi_{3/2}^S + i\delta_{I=1}^S) \quad (5)$$

$$\bar{P} = P_{1/2}\exp(-i\xi_{1/2}^P + i\delta_{I=1}^P) + \frac{1}{2}P_{3/2}\exp(-i\xi_{3/2}^P + i\delta_{I=1}^P). \quad (6)$$

To derive Eq. 4 and 5 one uses also definitions

$$\alpha := \frac{2\text{Re}(S^*P)}{|S|^2 + |P|^2} \quad \text{and} \quad \beta := \frac{2\text{Im}(S^*P)}{|S|^2 + |P|^2} \quad (7)$$

The ratios of the $\Delta I = 3/2$ to $\Delta I = 1/2$ amplitudes: $s_3 := S_{3/2}/S_{1/2}$ and $p_3 := P_{3/2}/P_{1/2}$ are the small parameters to expand the solution in the Maclaurin series. The expansion up to the next order (including the first not included term $\Delta_{3/2}$) for Eq. 4 and 5 reads:

$$\frac{\alpha + \bar{\alpha}}{\alpha - \bar{\alpha}} = -\tan(\delta_{I=1}^P - \delta_{I=1}^S) [\sin(\xi_{1/2}^P - \xi_{1/2}^S) + \Delta_{3/2}] \quad (8)$$

$$\frac{\beta + \bar{\beta}}{\alpha - \bar{\alpha}} = \tan(\xi_{1/2}^P - \xi_{1/2}^S) + \Delta_{3/2} \quad (9)$$

$$\Delta_{3/2} = s_3 \sin(\xi_{1/2}^S - \xi_{3/2}^S) - p_3 \sin(\xi_{1/2}^P - \xi_{3/2}^P). \quad (10)$$

14. Line 71: Could you elaborate how is this observable complementary to ϵ' (besides that fact that it contains only an s-wave)?

Response: (This is the combined answer for comment 11 and comment 14 from Reviewer 2 and comment 4 from Reviewer 3.) By summarising the conclusions from the theory papers on CP violations, we find that hyperon measurements can be considered complementary to ϵ'/ϵ measurement in kaons in two aspects:

- Direct CP violation effects in kaon decays must involve both isospin transitions $\Delta I = 1/2$ and $\Delta I = 3/2$, where the CP-odd phases come from QCD [Fig. 1(c)] and electroweak [Fig. 1(d)] penguins, respectively. There is a delicate balance between the two contributions and only if they are equal in size, they cancel exactly.
- In hyperon decays, the CP-violation signal comes predominantly from $\Delta I = 1/2$ transitions involving the QCD penguins [Fig. 1(c)].

Note that the schematic diagrams in Fig. 1 are the same for SM and BSM contributions. The only difference is that in SM, we know that the "blob" must contain a loop involving u , c and t quarks whereas a BSM scenario would involve contributions from hitherto unknown particles. From the aforementioned comparison of direct CP-violation in kaon and hyperon decays, one sees that the latter will be sensitive to the effects generated by the QCD penguin operators. Moreover, due to the vector nature of gluon exchanges, one expects a non-trivial helicity structure of the sdg vertex. In particular, certain BSM mechanisms might only contribute to hyperon decays. One such scenario, involving chromomagnetic penguin operators, is discussed by J. Tandean in ref. . This illustrates the second aspect of the complementarity – the sensitivity to additional CP-violation mechanisms. We realise that this discussion must be summarised in an elementary way in the manuscript. We propose to include the following sentence, L200-203:

The contributions to ϵ and ϵ' from hyperon decays on the one hand and kaon decays on the other, are described by different combinations of quark operators. In addition, hyperons provide information on the spin structure of the operators that is not possible to obtain from kaon decays.

15. Line 101: Add references.

Response: We added these references:

- Polarization studies: Klempt et al, Phys. Rep 368, p 119 (2002),
- Polarization at the LHC: ATLAS Coll., Phys. Rev. D 91, 032004 (2015).
- Spectroscopy: Phys Rev C 93, 065201 (2016).
- Heavy ions: STAR Coll. Nature (London) 548, 62 (2017).
- Decay properties of heavier baryons (example here Omega): HyperCP Coll: Phys Lett B 617, 11 (2005) and Phys Rev D 71, 051102 (2005).

Figure 1. Topology of the four-quark operators providing contributions to the CP-odd phases in weak decays of hadrons with strange quark

16. Line 105: the notation is confusing and gives the impression that cc applied only to the Lambda decay.

Response: We have modified the sentence "...and the corresponding \bar{E}^+ chain."

17. Line 144: How are the remaining background events treated in the likelihood fit?

Response: In the nominal results, the likelihood fit was performed without subtracting the background events as its effect on the final results are small. We performed a cross check confirming this by using the side-band subtraction method. In this method, two regions are selected: one below and one above the signal peak region. The side-band regions are equal in width to the signal region (indicated by the blue lines in Figure 2, (supplementary)). As the side-band events should have the same features as the background under the signal peak, they can be subtracted in the likelihood fit. The relative difference between the nominal and background subtracted results is small. Discrepancies smaller than 0.5% were observed for all parameters except for $\phi_{\bar{E}}$ (5%) and $\phi_{\bar{E}^+}$ (3%). These differences are however small in absolute numbers, $\delta\phi = 6 \cdot 10^{-4}$ and $\delta\bar{\phi} = 7 \cdot 10^{-4}$. One could consider taking the (absolute) difference between the nominal and background subtracted results as an additional systematic uncertainty. However, doing this results in a nearly unchanged Table 1 (the one change would be the systematic uncertainty of $\Delta\Phi$, changing from 0.016 to 0.017). Though the background will be of importance for future high-precision measurements, it has a negligible impact in this study.

We added a sentence about the background in the supplementary section (Methods), L302-303.

18. Line 249: How is the simulation adjusted?

Response: In the configuration file with the particle properties, the input mass of E is shifted by +95 keV/c² above the PDG tabulated value. We changed the word "value" to "input mass value" to make it more clear.

19. Line 251: Can you give an estimate of how large are the simulation samples.

Response: The following Monte Carlo samples have been used:

- 19.6×10^6 generated $J/\psi \rightarrow E^- \bar{E}^+ \rightarrow \Lambda \pi^- \bar{\Lambda} \pi^+ \rightarrow p \pi^- \pi^- \bar{p} \pi^+ \pi^+$ phase space. This sample was used for calculating the normalization in the maximum log likelihood method. The sample size was chosen in such a way that the number

of simulated and reconstructed events amounts to approximately 35 times the size of the experimental data sample.

- 5.74×10^6 generated $J/\psi \rightarrow \Xi^- \bar{\Xi}^+ \rightarrow \Lambda \pi^- \bar{\Lambda} \pi^+ \rightarrow p \pi^- \pi^- \bar{p} \pi^+ \pi^+$ events simulated using the parameters estimated from data, as input. This was used as a control sample searching for inconsistencies between data and MC and used for input/output checks. The sample size was chosen in such a way that the number of simulated and reconstructed events amounts to approximately 10 times the size of the experimental data sample
- 1200×10^6 J/ψ inclusive sample is used to study potential background contributions. The J/ψ are produced by the KKMC generator [1] and the known decay modes were generated with *EvtGen* [2], while the unknown decays were generated by *LundCharm* [3].
- 5.0×10^5 events of $J/\psi \rightarrow \gamma \eta_c (\rightarrow \Xi \bar{\Xi})$
- 5.0×10^5 events of $J/\psi \rightarrow \Sigma^{*-} \bar{\Sigma}^{*+} \rightarrow \Lambda \pi^- \bar{\Lambda} \pi^+$
- 2.5×10^5 events of $J/\psi \rightarrow \Xi^{*-} \bar{\Xi}^{*+} \rightarrow \pi^0 \Xi^- \bar{\Xi}^+$ and 2.5×10^5 events of the charge conjugated channel.

Response: The in-depth studies of the three latter samples were motivated by the comments from the referees.

20. Line 260: Though the cuts seem to select very pure samples is there any mis-identification that remain?

Response: Our Monte Carlo studies show that the probability of mis-identifying a proton(anti-proton) for a π^+ (π^-) is 0.17% (0.18%). We included the mis-identification rate in the updated draft, L268-269.

21. Line 269: Are there multiple candidates? If yes how are they handled?

Response: Our signal Monte Carlo studies show that the combination of in total six tracks in each Ξ^- and $\bar{\Xi}^+$ chain results in multiple candidates. After the momentum selection all proton(anti-proton) and π^- (π^+) candidates are combined and subjected to vertex fits. The vertex fit finds a common decay vertices found from the $p - \pi$ and $\Lambda - \pi$ track combinations, and updates the momentum information. The main objective of the vertex fit is to improve the resolution of the Λ and Ξ invariant mass. This is done by performing the vertex fit in two steps. In the first step a common decay vertex is found from the $p - \pi$ and $\Lambda - \pi$ track combinations. In the second step the fit also includes information about the Λ and Ξ production point. With the improved resolution we select the best candidate from the combination which minimises $((m_{p\pi} - m_{\Xi})^2 + ((m_{p\pi} - m_{\Lambda})^2)$. This analysis algorithm is efficient of finding the correct combination. We investigated how often a pion from a $\Xi^- (\bar{\Xi}^+)$ decay is wrongly assigned to be the $\pi^- (\pi^+)$ from the $\Lambda (\bar{\Lambda})$ decay and vice versa. The probability π^+ being wrongly assigned was found to be 0.51%. The corresponding probability for π^- is 0.49%.

22. Line 273: Does the kinetic fit require that the Lambda mass is set to the PDG value?

Response: No mass constraint for Λ and $\bar{\Lambda}$ is included. The fit hypothesis is for the reaction $e^+e^- \rightarrow J/\psi \rightarrow \Xi^- \bar{\Xi}^+$, where the reconstructed Ξ^- and $\bar{\Xi}^+$ four-momenta obtained from vertex fits are used without the particle mass hypotheses.

23. Line 276: What fraction of the background mentioned here remains in the analysis?
Response: After kinematic fit the $J/\psi \rightarrow \gamma\eta_c(\rightarrow \Xi\bar{\Xi})$ remaining fraction is 0.23%. For $J/\psi \rightarrow \Xi^*\bar{\Xi}^+ \rightarrow \pi^0\Xi^-\bar{\Xi}^+$ (or c.c.) the remaining fraction is less than 10^{-5} .
24. Line 290: Does this requirement cost a lot of signal efficiency?
Response: No, it does not. We loose 2.5% of the events when imposing $|\cos\theta_{\Xi,cm}| < 0.84$ compared to having no requirement on $\cos\theta_{\Xi,cm}$.
25. Line 291: Could you add a plot to illustrate the final set of data used for the analysis?
Response: The final event sample is shown in the right panel, Figure (supplementary) 2. We missed to refer to it in the Methods text. We now included the text "This is shown in the right panel of Figure 2 (supplementary)", L301.
26. How is the number of signal evaluated ? Do you make the assumption that there is no background remain at this stage?
Response: The number of event candidates are those remaining after all selection criteria have been applied. In this event sample there is a small contamination of background, estimated to be 199 ± 17 events (0.27%). A cross-check which takes into account this background shows that the background can be neglected without affecting the determined parameters. See our response in comment 17. For the updated number of background events, see comment 3. We added further text in Methods (L300-303):
After applying all aforementioned selection criteria, 73 244 $\Xi^-\bar{\Xi}^+$ candidates remain in the final sample. This is shown in the right panel of Figure 2 (supplementary). The number of remaining background events are estimated to be 199 ± 17 . The background contribution has a marginal effect on the results at this precision and is therefore neglected.
27. Line 295: Can you provide more information about the efficiency? How it's computed? How do you insure that it does not distort the results? The treatment of the efficiency does not seem to enter in the lists of systematic uncertainties?
Response: The efficiency is determined from the full MC simulation of the detector response. The information about efficiencies comes from the common BESIII particle reconstruction studies and is included in the MC program. The final cross checks, specific for our analysis, are carried out using our simulated and experimental data. Systematic effects 2-6 in the list on pages 22-24 are all related to uncertainties in the efficiency determination, studied for each selection criterion.
28. Line 315: Do you have an explanation why the product agrees with the previous result?
Response: The method for measuring the product $\alpha_\Lambda\alpha_\Xi$ used by the E756 collaboration is scientifically sound, and to first order the value of α_Λ does not enter explicitly. We therefore expect to obtain a result that agrees with theirs, and find it reassuring that it does. The strength of our method in this case is that we can determine α_Ξ and α_Λ separately.
29. Line 335: are the uncertainties also well behaved from this procedure?
Response: Yes, they are in agreement with the experimental fit uncertainties. We included "and uncertainties" in the updated manuscript.
30. Line 337: Does the variation of the kinetic fit impact the background contamination ? If yes how this is taken into account in each of the 20 steps ? Same question for the Lambda and Xi mass windows systematic uncertainties.
Response: The relative background contribution stays nearly the same when varying the χ^2 of the kinematic fit and the mass windows of Λ/Ξ . If one would require $\chi^2 < 200$

instead of the nominal $\chi^2 < 100$, the non-peaking background contribution is 0.27% (204 events). The peaking contribution from $J/\psi \rightarrow \gamma \eta_c \rightarrow \bar{E} E$ changes by a few events from already low levels (12 ± 5 for main selection). Hence far too little to impact the results. Similar low background levels are seen for tighter selections. For the determination of the uncertainties induced by the Λ and E mass windows, the relative background is approximately 0.25% or lower. Hence the background plays a negligible role.

31. Figure 1: are these distribution efficiency corrected?

Response: None of the supplementary figures show efficiency corrected distributions. The integrated counts in Figure 1 (supplementary) is six times that of the final event sample, as there is an entry for each particle of the six final state particles.

32. Figure 1/2 of supplementary material: are both run periods included?

Response: Yes, both run periods are included.

IV. Referee 3

Dear authors,

The paper presents the first application, using the BESIII data, of a very sensitive test of CP symmetry, in decays of doubly strange hyperons (“Xi”), to test the Standard Model and search for new physics, in the same spirit as the measurements of ϵ'/ϵ in kaons. The method was presented before in references [2] and [26] of the paper draft in 2019, where a sensitivity study had been done, but this paper shows the first application of this method in an analysis of experimental data. The method is based on a complex angular analysis exploiting the spin correlations of the pairs of $E-\bar{E}$ produced in the decays of J/ψ made at the BESIII experiment, in their weak decay chains ($E \rightarrow \Lambda\pi; \Lambda \rightarrow p\pi$). This angular analysis allows one to disentangle the P-wave from the S-wave contributions, and measure the complete set of decay parameters. The novelty in the results is the use of this new method on real data, which allows not only very precise CP symmetry tests (improvement of the sensitivity of an order of magnitude), but also to disentangle the weak – (or new physics) parts from the strong interaction parts, and get more precise and direct measurements. This is complementary to the ϵ'/ϵ measurement in kaons, which has uncertainties from strong interactions. The accuracy of the results is dominated by the statistical uncertainty, and the accuracy gets close to the one of the theoretical prediction from the standard model. So even if CP violation in E decays is not observed, the measurement is a very precise test of the standard model and paves the way for future experiments like PANDA at FAIR which could reach even better sensitivity, and see new physics if it exists. New parameters on E decays have also been measured for the first time, and can be useful for other studies like spectroscopy. Parameters on Λ decays are also measured very precisely and with a more direct approach, and can be compared to results of other experiments.

Those results deserve being published here: this first application on data of a very promising method, is much more accurate, more direct, and more complete with different “views” (like a tool box) of CP tests compared to the classical CP asymmetry approach. It also provides measurements of interesting new parameters, and a fundamental test of the Standard Model. There are also interesting and precise additional results on the Λ hyperon that can be compared to other experiments, and the E has been studied more deeply.

The methodology is well described (selection of data, observables, likelihood fit, extraction of the parameters), looks robust and it has been validated with full Monte-Carlo simulations (an order of magnitude bigger in statistics compared to the data). The data have clear signals and

relatively low backgrounds. Systematic effects are studied in details. The analysis seems robust.

There are a couple questions to clarify (see below). The paper is well written, the references are appropriate, the abstract and summary, introduction and conclusion are clear, even if the analysis is complex. A few improvements are suggested below.

Suggested improvements and questions:

1. Data fit Quality test for the likelihood fit procedure: did you compare the Chi2 value at the maximum likelihood you obtain with the data, to the distribution of the Chi2 values obtained for the 10 subsamples of the Monte-Carlo (same statistics as data, and parameters close as the ones found in the data), in order to check the fit “quality” in the data (cf lines 330 to 335)?

Response: The goodness-of-fit for the maximum log likelihood method is quantified by the negative log-likelihood value, mlv . The experimentally obtained value is $mlv_{EXP} = -5439.3$ while the quality test using the 10 MC sub-samples gives $mlv_{MC} = -5427.5 \pm 117.7$, hence in good agreement. In addition, the uncertainties and correlations are reproduced by the bias test.

2. Systematics (cf lines 321 to 327) – Does it mean if you think according to that criteria described there that the shift is due to a statistical fluctuation, you don’t account for that as a systematics? If you had, all the effects added up would have been big compared to the “real” systematics effects? Is that method to evaluate if we have a real systematics described in reference [47,48] fully reliable? Is the cut at 2 sigma optimized?

Response: Yes, you got this correctly. We perform several consistency checks and if we observe significant differences compared to the nominal values, we evaluate the systematic uncertainties. If the differences are not significant, they are likely due to statistical fluctuations rather than true systematic effects. Including a systematic uncertainty for every such fluctuation would lead to an overestimation of the uncertainties. The approach that we use are outlined e.g. in the BaBar document on Recommended Statistical Procedures [6] and is considered well-established in precision experiments. The criterion of a fluctuation larger than two sigma for a test to fail, is related to the number of checks we perform which is approximately ten for each cut. From the definition of standard deviations, we expect one out of twenty fluctuations to occur outside the two sigma limit. If one out of ten checks is outside, then there is a non-negligible chance that it is not a statistical fluctuation but a true effect. With a limit of one sigma, we expect three out of ten tests to “fail”, hence a limit of one sigma would result in a large number of statistical fluctuations being misinterpreted as systematic effects. This in turn would inflate the total systematic uncertainty.

3. Xi decay length distribution The decay length can vary, so it is accounted for in the likelihood fit, but not considered as an “interest” parameter. What is the typical average decay length (the lifetime is close to 100 ps)? Have you checked the decay length distribution to be consistent with the Xi decay, without carefully re-measuring the Xi lifetime, but as a test?

Response: The Monte Carlo sample is generated with the established life-time of the Ξ and is in good agreement with data. We also tested the consistency by varying the decay length selection criteria (see L367-370) and we found no systematic effect.

4. It is said in the conclusion that this measurement is “complementary” to epsilon’/epsilon measurement in kaons. What do you mean exactly? It is the same spirit, looking for CP violation in strange hadrons decays, to test the standard model and look for beyond

standard model physics. The method is different, as the type parameters studied, so the “view” on the CP tests (also clear of strong phase effects), and the strange hadron too... It is complementary in the method and approach or could it shed a different light on possible new physics? Have you examples of possible new physics scenarios that would affect differently the measurements of ϵ'/ϵ and this analysis?

Response: Answer to this question is now combined with the answer to Reviewer 2, question 14.

5. Line 155 “allows for three independent CP symmetry tests”. It seems those tests are detailed from lines 156 to line 178. But they are not clearly listed as first test, second test, third test. You can guess that (first test, classical : A_{CP} , second test $\Delta\phi_{CP,\mathcal{E}}$, third $\xi_P - \xi_S$ weak phases differences, with the last two tests very new) ? But it is not fully highlighted. Also in table (supplementary) 3, systematics uncertainties on the CP tests, in the caption it would be good to remind the reader of the 3 different CP tests with the corresponding parameters (1: $A_{CP,\Lambda}$ & $A_{CP,\mathcal{E}}$; 2: $\Delta\phi_{CP,\mathcal{E}}$; 3: weak phase difference but you need to fit also the strong phases together with it)

Response: The three independent tests are $A_{CP,\Lambda}$, $A_{CP,\mathcal{E}}$ and $\Delta\phi_{CP,\mathcal{E}}$. The weak phase difference is a CP test, but not independent since it is calculated from $\langle \alpha_{\mathcal{E}} \rangle$ and $\Delta\phi_{CP,\mathcal{E}}$. In the present manuscript version, we clarify this in the introductory paragraph:

"We present the first direct determination of the weak phase difference for any baryonic decay: $(\xi_P - \xi_S) = (1.2 \pm 3.4 \pm 0.8) \times 10^{-2}$ rad. From our measured decay parameters, three independent CP observables can be calculated: $A_{CP}^{\mathcal{E}}$, $\Delta\phi_{CP}^{\mathcal{E}}$ and A_{CP}^{Λ} . The former two observables are measured for the first time and found to be $A_{CP}^{\mathcal{E}} = (6 \pm 13 \pm 6) \times 10^{-3}$ and $\Delta\phi_{CP}^{\mathcal{E}} = (-5 \pm 14 \pm 3) \times 10^{-3}$ rad, respectively."

6. For the results for which there was a previous measurement by another experiment (lots of results are also first measurements), and for which there is a “tension”, but the approach is different, are there plans for more studies or hints of why it could be so?

Response: One of the most important tensions is the value of the $\phi_{\mathcal{E}}$ parameter. Such measurement requires an intensive source of polarised \mathcal{E} or spin-entangled $\mathcal{E}^- \bar{\mathcal{E}}^+$ systems. It might be considered at Super-charm τ , PANDA and Belle-II.

7. a. Line 69: why is the strong phase difference between P and S the ratio beta over alpha?

Response: This relation is derived e.g. in J. Donoghue and S. Pakvasa (Ref 22 in the current manuscript version). Here we present the main arguments. The amplitude for the $\mathcal{E} \rightarrow \Lambda\pi$ decay can be described as

$$A(\mathcal{E}^- \rightarrow \Lambda\pi^-) = S + P\sigma \cdot \hat{\mathbf{n}}, \quad (11)$$

where $\hat{\mathbf{n}} = \mathbf{q}/|\mathbf{q}|$ is the direction of the Λ momentum \mathbf{q} in the \mathcal{E} rest frame. The S and P amplitudes can be written in the isospin basis as

$$S = S_{1/2} \exp(i\xi_{1/2}^S + i\delta_{I=1}^S) + \frac{1}{2} S_{3/2} \exp(i\xi_{3/2}^S + i\delta_{I=1}^S) \quad (12)$$

$$P = P_{1/2} \exp(i\xi_{1/2}^P + i\delta_{I=1}^P) + \frac{1}{2} P_{3/2} \exp(i\xi_{3/2}^P + i\delta_{I=1}^P), \quad (13)$$

where $S_{\Delta I}$ ($P_{\Delta I}$) and $\xi_{\Delta I}^S$ ($\xi_{\Delta I}^P$) are the magnitude and the CP-odd phase for the two possible weak transitions changing isospin I by $\Delta I = \frac{1}{2}$ and $\frac{3}{2}$, respectively. The phase shifts $\delta_{I=1}^S$ and $\delta_{I=1}^P$ represent the strong π - Λ scattering in the final state. They are labelled by the isospin $I = 1$ value in the final state. The magnitudes $S_{\Delta I}$ and $P_{\Delta I}$ are real numbers. To evaluate

β/α , we set the weak CP-odd phases $\xi_{\Delta I}^S(\xi_{\Delta I}^P)$ to zero and use definitions of the α and β parameters:

$$\alpha = \frac{2Re(S^*P)}{|S|^2+|P|^2} = \frac{2(S_{1/2}+\frac{1}{2}S_{3/2})(P_{1/2}+\frac{1}{2}P_{3/2})}{|S|^2+|P|^2} \cos(\delta_{I=1}^P - \delta_{I=1}^S) \quad (14)$$

and

$$\beta = \frac{2Im(S^*P)}{|S|^2+|P|^2} = \frac{2(S_{1/2}+\frac{1}{2}S_{3/2})(P_{1/2}+\frac{1}{2}P_{3/2})}{|S|^2+|P|^2} \sin(\delta_{I=1}^P - \delta_{I=1}^S); \quad (15)$$

and therefore

$$\frac{\beta}{\alpha} = \tan(\delta_{I=1}^P - \delta_{I=1}^S). \quad (16)$$

In the first version of the manuscript, we used term "leading order" when discussing the extraction of the weak phase difference. This means we neglect $S_{3/2}/S_{1/2}$ and $P_{3/2}/P_{1/2}$ terms which are of the order of 10%.

8. b. Line 251: Monte-Carlo parameters close to the parameters measured in data (table 1). It means that initially those values were unknown, and once they were known, the simulation is reweighted or redone, so is it kind of an iterative process?

Response: In order to compare distributions and perform general consistency checks, we produced an independent Monte Carlo sample that was generated with parameter values obtained with real data as input. However, this is not a true iterative procedure, since the updated Monte Carlo generation is only performed once. To avoid possible confusion, we would like to stress that in order to determine the parameter values in the maximum-log-likelihood fit, a **phase space** simulated Monte Carlo sample must be used for the normalisation (phase space means that all parameter values are set to 0).

9. c. Line 290: is the cut on the Xi polar angle optimized for 0.84? If yes, how?

Response: When comparing the 1D $\cos\theta$ distribution for charged tracks, there is a difference between experimental data and Monte Carlo data at large polar angles. This effect becomes non-negligible at $\cos\theta \approx 0.84$ and results in a systematic bias on the parameter values. By imposing $|\cos\theta_{\Xi,CM}| < 0.84$, the effect is reduced to a negligible level. The cost of this criterion is low - the final event sample is reduced by only 2.5%. See also comment 24.

10. Line 344 - "mass windows are investigate" -> investigated?

Response: Corrected.

11. Line 351 "lower than to the main" -> lower than the main?

Response: Corrected

V. Additional changes

There have been a few more updates to the manuscript which requires further explanation.

1. We have tried to make it clearer that the calculation of the strong phase were made with the average values $\langle \phi_{\Xi} \rangle$ and $\langle \alpha_{\Xi} \rangle$ as input (line 173) and the calculation of the weak phase used $\langle \alpha_{\Xi} \rangle$ and $\Delta\phi_{\Xi}$ as input (lines 79-80).
2. We have reduced the number of significant digits in such a way that it matches the size of the uncertainty for the CP asymmetries A_{CP}^{Ξ} , $\Delta\phi_{CP}^{\Xi}$ and A_{CP}^A .
3. Motivated by a comment from Referee 2, we carried out further studies on the background contributions. Although these studies did not change our results, we want to make an

adjustment to the background contribution. We found that while most of the background is continuous, there exists a small peaking contribution from the channel $J/\psi \rightarrow \gamma \eta_c (\rightarrow \bar{E}E)$, 12 ± 5 events. Furthermore, we found that $J/\psi \rightarrow E^* \bar{E}^+ \rightarrow \pi^0 E^- \bar{E}^+$ (or c.c.) does not contribute to the final event sample. Therefore we changed the estimated number of background events from 187 ± 16 events to 199 ± 17 events.

4. Modification of caption text, Figure (supplementary) 2.
5. Small corrections made to some of the references.

REFERENCES

- [1] S. Jadach, B. F. L. Ward and Z. Was, *Comp. Phys. Commu.* **130**, 260 (2000); *Phys. Rev. D* **63**, 113009 (2001).
- [2] R. G. Ping *et al.*, *HEP&NP*. 32 (2008) 599.
- [3] J. C. Chen *et al.*, *Phys. Rev. D* 62, 034003 (2000).
- [4] J. F. Donoghue, X. G. He and S. Pakvasa, *Phys. Rev. D* **34** (1986), 833.
- [5] J. Tandean, *Phys. Rev. D* **69** (2004), 076008.
- [6] BaBar *Recommended Statistical Procedures*, BABAR Analysis Document no. 318, Version 1.

Reviewer Reports on the First Revision:

Referees' comments:

Referee #2 (Remarks to the Author):

Dear Authors,

Thank you for addressing my comments and questions in a detailed manner. I have two minor remaining comments.

- "The contributions to ε and ε' from hyperon decays on the one hand and kaon decays on the other, are described by different combinations of quark operators. In addition, hyperons provide information on the spin structure of the operators that is not possible to obtain from kaon decays. "

Thank you for adding a sentence. Discussing operators here suggests the usage of an EFT of kind ? if Yes what kind of EFT ? "The spin structure of operators" is unfortunately unclear for non-experts. Could you try to clarify/extend this ?

The quality of Figure 2 looks much better, thank you. Please use the same style for the figures in the rest of the text ?

Referee #3 (Remarks to the Author):

Dear Authors,

I am satisfied with the answers to my comments, as well as with the improvements in the manuscript. I still think this manuscript deserves to be published here. Even if no CP violation has been observed yet, the use for the first time on data of this new method is paving the way and demonstrates very nice possibilities for the future experiments. This sophisticated method is not only very accurate, but it also offers a new view on the direct CP violation studies with light strange hadrons, which is complementary to the studies of kaons.

I have also checked the questions and answers to the other reviewers

Here are my more detailed answers to the authors responses.

Question 1 (Reviewer 3)

That is perfect, and it is important to have done this check

Question 2 (Reviewer 3)

I guess it makes sense, May-be the choice of exactly 2 sigmas is not what would have been optimal (something between 1 and 2 sigmas would have worked too?) But the important thing is not to underestimate the systematics as well. I guess the studies done in the reference checked that the

choice of 2 sigmas is conservative.

Question 3 (Reviewer 3)

My question was : did you try to fit the lifetime on data as a cross-check, but this is not really necessary, the check you have done is sufficient.

Question 4 (Reviewer 3)

Thank you for that detailed answer, this question is important as the study on direct CP violations using kaons has been an important program. It seems it is not so simple to explain (probably that is why you did not do it in the first place). But you managed to do it in a few words added to the manuscript.

This was important.

Question 5 (Reviewer 3)

Thank you very much for the clarification and the appropriate change in the manuscript

Question 6 (Reviewer 3)

Thank you for the response

Question 7 (Reviewer 3)

Thank you very much for the explanation. I understand better.

It is apparently not so simple to explain shortly. I guess the references are enough.

Question 8 (Reviewer 3)

Thank you for the clarification. I guess the procedure you are using is correct. In some other experiments, a "reweighted MC" is used for different values of the parameters without having to regenerate the MC which is time consuming. But th procedure you are using is correct as well.

Question 9 (Reviewer 3)

I guess that is a correct way to do. But do you have any understanding where this difference between data and Monte-Carlo comes from? It would be reassuring.

I am happy with the changes and improvements in the text and the references.

Author Rebuttals to First Revision:

Comments to Referees, second iteration:
Weak phases and CP-symmetry tests in sequential decays of entangled double-strange baryons
Nature manuscript 2021-05-08135

BESIII collaboration

Referee 2

Dear Authors, Thank you for addressing my comments and questions in a detailed manner. I have two minor remaining comments. General comments and questions :

1. *The contributions to ϵ and ϵ' from hyperon decays on the one hand and kaon decays on the other, are described by different combinations of quark operators. In addition, hyperons provide information on the spin structure of the operators that is not possible to obtain from kaon decays.*

Thank you for adding a sentence. Discussing operators here suggests the usage of an EFT of kind ? if Yes what kind of EFT ?

Response: This is EFT of low energy weak interactions where the W and Z propagators are replaced by a set of local operators multiplied by elements of Cabibbo-Kobayashi-Maskawa matrix and the Wilson Coefficients. A subset of this EFT is Fermi theory of beta decays.

2. *The spin structure of operators is unfortunately unclear for non-experts. Could you try to clarify/extend this ?*

Response: The local four-quark operators of the EFT can include non-trivial color rearrangements corresponding to the exchange of gluons (spin 1 bosons). The contribution of such operators will depend on spin orientation of the quarks. Therefore in processes where the baryons are polarized the contribution of the respective operators might be different. We have discussed one such example related to chromo-magnetic operators.

3. *The quality of Figure 2 looks much better, thank you. Please use the same style for the figures in the rest of the text ?*

Response: We have now updated the supplemental figures as well. If there are further changes we will co-operate with the layout experts at Nature.

Referee 3

Dear Authors,

I am satisfied with the answers to my comments, as well as with the improvements in the manuscript. I still think this manuscript deserves to be published here. Even if no CP violation has been observed yet, the use for the first time on data of this new method is paving the way and demonstrates very nice possibilities for the future experiments. This sophisticated method is not only very accurate, but it also offers a new view on the direct CP violation studies with light

strange hadrons, which is complementary to the studies of kaons. I have also checked the questions and answers to the other reviewers Here are my more detailed answers to the authors responses.

4. That is perfect, and it is important to have done this check.
5. I guess it makes sense, May-be the choice of exactly 2 sigmas is not what would have been optimal (something between 1 and 2 sigmas would have worked too?) But the important thing is not to underestimate the systematics as well. I guess the studies done in the reference checked that the choice of 2 sigmas is conservative.
6. My question was : did you try to fit the lifetime on data as a cross-check, but this is not really necessary, the check you have done is sufficient.
Response: No we did not perform this fit to measure the life time. As it was not the purpose nor relevant for our study and would require other types of cross checks.
7. Thank you for that detailed answer, this question is important as the study on direct CP violations using kaons has been an important program. It seems it is not so simple to explain (probably that is why you did not do it in the first place). But you managed to do it in a few words added to the manuscript. This was important.
8. Thank you very much for the clarification and the appropriate change in the manuscript.
9. Thank you for the response.
10. Thank you very much for the explanation. I understand better. It is apparently not so simple to explain shortly. I guess the references are enough.
11. Thank you for the clarification. I guess the procedure you are using is correct. In some other experiments, a "reweighted MC" is used for different values of the parameters without having to regenerate the MC which is time consuming. But the procedure you are using is correct as well.
12. I guess that is a correct way to do. But do you have any understanding where this difference between data and Monte-Carlo comes from? It would be reassuring.
Response: There are only charged final state particle tracks in this analysis. Therefore the small differences which we see is most likely attributed to a different magnetic field map description in simulation compared to data.

I am happy with the changes and improvements in the text and the references.